# A Photosensitized Singlet Oxygen (^1^O_2_) Toolbox for Bio-Organic Applications: Tailoring ^1^O_2_ Generation for DNA and Protein Labelling, Targeting and Biosensing

**DOI:** 10.3390/molecules27030778

**Published:** 2022-01-25

**Authors:** Dorien Aerssens, Enrico Cadoni, Laure Tack, Annemieke Madder

**Affiliations:** Organic and Biomimetic Chemistry Research Group, Department of Organic and Macromolecular Chemistry, Ghent University, Krijgslaan 281-S4, 9000 Gent, Belgium; Dorien.Aerssens@UGent.be (D.A.); Enrico.Cadoni@UGent.be (E.C.); Laure.Tack@UGent.be (L.T.)

**Keywords:** singlet oxygen, bioorganic, DNA targeting, protein targeting, photosensitizer conjugates, oxidative damage, ROS, CALI, mini-SOG, biosensing

## Abstract

Singlet oxygen (^1^O_2_) is the excited state of ground, triplet state, molecular oxygen (O_2_). Photosensitized ^1^O_2_ has been extensively studied as one of the reactive oxygen species (ROS), responsible for damage of cellular components (protein, DNA, lipids). On the other hand, its generation has been exploited in organic synthesis, as well as in photodynamic therapy for the treatment of various forms of cancer. The aim of this review is to highlight the versatility of ^1^O_2,_ discussing the main bioorganic applications reported over the past decades, which rely on its production. After a brief introduction on the photosensitized production of ^1^O_2_, we will describe the main aspects involving the biologically relevant damage that can accompany an uncontrolled, aspecific generation of this ROS. We then discuss in more detail a series of biological applications featuring ^1^O_2_ generation, including protein and DNA labelling, cross-linking and biosensing. Finally, we will highlight the methodologies available to tailor ^1^O_2_ generation, in order to accomplish the proposed bioorganic transformations while avoiding, at the same time, collateral damage related to an untamed production of this reactive species.

## 1. Introduction

In the past few decades, the ‘activated’ form of oxygen, singlet oxygen (^1^O_2_), has been intensely reviewed for its use as an oxidation reagent in several fields of science, ranging from pure synthetic organic chemistry to nanoscience, encompassing medical and pharmaceutical applications as well [1,2,3,4,5,6,7,8]. Generation of ^1^O_2_ from ground state triplet oxygen can occur in a large variety of manners. Examples include the reaction of hydrogen peroxide in the presence of hypochlorite, or the thermal denaturation of naphthalene endoperoxides [9,10]. Its production has been demonstrated in a cellular environment, catalyzed by peroxidase enzymes, notably myeloperoxidase and eosinophil peroxidase, which play prominent roles in the inflammatory mechanisms [11].

Remarkably, the photosensitized production of ^1^O_2_, only requiring the presence of a photosensitizer (PS) and light as an external stimulus, has gained considerable significance in various fields of chemistry [8,12], as it conveniently allows spatiotemporal control of the oxygen activation. In the 1960s, it was proposed by C. S. Foote and S. Wexler that ^1^O_2_, the excited state of molecular oxygen with the lowest energy, was the reactive intermediate in photosensitized oxidation reactions. This proved to be a milestone in ^1^O_2_ mediated research, paving the way for its use in various (bio)organic reactions and photodynamic therapy [13,14,15,16]. A simple and visual explanation for ^1^O_2_ generation can be provided by following the so-called Jablonski diagram (Figure 1). The diagram shows the process of intersystem crossing, a weakly allowed pathway in which an excited singlet state PS is converted to its excited triplet state (PS*) [17]. The light-dependent photo-oxidation mechanisms can be of two different types, based on the two main pathways by which the energy of the PS (PS*) is transferred to molecular oxygen. The type I mechanism encompasses electron transfer from the excited sensitizer to a nearby molecule (solvent, or other suitable substrates), leading to the formation of a radical species that, subsequently, reacts with molecular oxygen, causing the formation of oxidizing ROS, such as the superoxide radical [18]. The type II mechanism involves the direct interaction between the excited PS* and ground state oxygen [19]. According to the molecular orbital theory, triplet state oxygen (^3^Σ_g_^−^) contains two unpaired electrons occupying the two antibonding, degenerate orbitals (π*) with parallel spin. PS* can transfer energy to ground state ^3^Σ_g_^−^, promoting its excitation into a less stable, and high energy ^1^O_2_ state (^1^Δ_g_). Here, the two unpaired electrons occupy the same antibonding orbital with opposite spin. The formation of a second excited state with higher energy is also possible (^1^Σ_g_^+^), and only differs from ^1^Δ_g_ state by disposition of the electrons, which occupy both antibonding orbitals of the molecular oxygen. However, the latter readily relaxes to ground state in aqueous media with a short lifetime and plays a less important role in this context [20].

Several light absorbing molecules were investigated for their ^1^O_2_ generation/photosensitizing capacity. In general, a decent PS should fulfill the following criteria, as described by M. C. DeRosa et al. The PS should exhibit sufficient photostability (1) and a high absorption coefficient (2) in the spectral area of the excitation light. The triplet state should possess the appropriate energy (3) and lifetime (4) to allow for efficient energy transfer to the ground state molecular oxygen. Additionally, an appropriate quantum yield ($\Phi_{T}$ > 0.4) of the triplet state (5) is required, meaning that a sufficient population of molecules, initially excited to the singlet state, should cross over to the triplet state [8]. The most common PSs can be categorized into several groups, such as organic dyes, porphyrins, transition metal complexes etc. Table 1 shows an overview of the different classes and gives some examples of PSs [8,21,22].

In organic synthesis, ^1^O_2_ has become an important sustainable green synthetic reagent as it is generated from air oxygen, natural dyes and visible light. Unlike ground state oxygen, it is a highly reactive, electrophilic and a non-radical molecule that can be involved in multiple reactions. Of high importance, is its nature as dienophile, for which ^1^O_2_ was extensively used in organic synthesis [23]. In ene reactions, it is used to produce allylic hydroperoxides from alkenes, whereas in [2 + 2]-cycloadditions with electron rich double bonds bearing no allylic protons and [4 + 2]-cycloadditions with 1,3-dienes, 1,2-dioxetanes and endoperoxides can be generated. More specifically of interest to us, especially in the context of bioorganic applications is the [4 + 2]-cycloaddition with furan. An outstanding review on the ‘powerful partnership’ between furans and ^1^O_2_ was published by Vassilikogiannakis et al. The mechanism of furan oxidation is shown in Figure 2 [24]. ^1^O_2_ is also known to oxidize sulphur, selenium, phosphorous and nitrogen compounds. Furthermore, superoxides can be formed as well via electron transfer reactions in which an electron is transferred from the electron-rich compound to the electrophilic ^1^O_2_.

^1^O_2_ is not only a powerful tool in organic synthesis. It also plays a crucial role in photodynamic therapy (PDT) [4]. Classically, PDT of tumors makes use of intracellular localization of PS, preferentially in tumor tissue. Upon light irradiation, ^1^O_2_ is generated, which can result in apoptosis and necrosis and thus the destruction of the cancerous cells. However, the widespread clinical use of PDT is hindered due to the limited tissue penetration of the excitation light. This is the Achilles’ heel of PDT treatment when treating deep-seated tumors under the skin [4]. Nevertheless, deeper penetration and less attenuation during tissue propagation can be achieved by making use of near infrared light or X-ray radiation. In addition, PSs can also be activated by internal self-luminescence to avoid problems such as limited tissue penetration. Furthermore, third generation PSs, where the PS is covalently attached to a tumor targeting moiety, have gained a lot of interest over the past few years due to their higher tumor selectivity [4,8]. This type of PS will be discussed in more detail in Section 4 of this review.

Throughout the last few decades, the number of applications involving ^1^O_2_ rose tremendously and the use of this reagent in a biological context should be evaluated closely since ^1^O_2_ is capable of efficiently reacting with unsaturated components of multiple cellular constituents. The oxidative damage on DNA, proteins and lipids is known to have detrimental effects on the cellular functioning, resulting in several diseases, such as arthritis and skin cancer [25]. In the first section of the present survey, the important ^1^O_2_-mediated oxidation pathways of different classes of biomolecules are briefly highlighted. However, a more detailed description is out of the scope of this review, since multiple excellent surveys on this topic are available [12,25,26,27]. Next, the biological applications of ^1^O_2_, such as its use in DNA cross-linking (CL), peptide and protein labelling and biosensing, are presented with attention for the possible collateral damage in those scenarios, referring to the oxidation pathways summarized in the first section. In the final section, improved strategies to tailor the ^1^O_2_ production towards a more contained, localized oxidation method (‘taming the bullet’) are discussed.

## 2. Reaction of Singlet Oxygen with Biomolecules

### 2.1. Peptides and Proteins

Peptides and proteins, built up by amino acids, are indispensable in biological systems. They are involved in a distinct range of vital cellular functions, for example in metabolic activity, catalysis, signal transduction, reading and translation of the genetic information (DNA). Additionally, proteins, such as collagen, create the structural framework of the cell [28]. As a consequence of their crucial role, oxidative damage, caused by reactive oxygen species (including ^1^O_2_), to this class of macromolecules can influence an entire biochemical pathway, finally resulting in the development of diseases.

In the early 1990s, the reactivity of ^1^O_2_ towards amino acids was evaluated in several quenching experiments [29]. The aromatic amino acids, tyrosine, tryptophan and histidine, the sulfur-containing methionine and cysteine and proline proved to be vulnerable to ^1^O_2_ [29].

^1^O_2_-mediated oxidation of cysteine can follow two pathways, whether type I or II reactions are occurring. In case of the former, cysteic acid and H_2_O_2_ are formed, while type II reactions give rise to the formation of cystine via a persulfoxide intermediate. The reaction mechanisms are not fully elucidated, but proposed reaction pathways are illustrated in Figure 3A [25,26,30,31]. The photo-oxidation of methionine is similar as seen for alkyl sulfides, so largely dependent on the reaction conditions (solvent). In protic solvents, the methionine oxidation causes the generation of a persulfoxide intermediate, leading to the final formation of two sulfoxide compounds (Figure 3B).

Early studies on the reactivity of ^1^O_2_ towards histidine describe a complex mixture of degradation products, but characterization of these intermediates was hampered due to stability issues [32,33]. Finally, Kang and Foote were able to identify the main endoperoxides via ^1^H and ^13^C-NMR analysis, elucidating the oxidation mechanism. First, ^1^O_2_ reacts via a [4 + 2] cycloaddition with the imidazole ring of histidine, giving a 2,5 endoperoxide. The latter rearranges to a hydroperoxide, which degrades to the corresponding alcohol (Figure 3C) [34].

Similarly, ^1^O_2_ oxidation of tyrosine proceeds through an endoperoxide intermediate, followed by the rearrangement into a labile hydroperoxide, as illustrated in Figure 3D. Finally, the respective alcohol is formed. The presence of a free amine in the amino acid leads to an intramolecular cyclisation, generating an isomeric mixture of cyclic peroxides, followed by a decay to the corresponding alcohol [35]. The indole-containing amino acid tryptophan can react with ^1^O_2_ via either a cycloaddition into a dioxetane or via an Alder-ene reaction to 3′-hydroxyindolenine. The preferred pathway will depend on both the sterical hindrance and the electron density of the reagents (more electron-rich indoles and sterically hindered ones will preferentially follow the cycloaddition pathway). The resulting unstable 3’-hydroxyindolenine undergoes an intramolecular addition of the amino side chain, giving rise to a hydroperoxypyrroloindole, which subsequently slowly decomposes to the corresponding alcohol. The dioxetane intermediate can either generate N-formylkynurenine as the expected ring cleavage product or can decompose through an initial O-O cleavage into dioxindolylalanine. The decomposition pathways are summarized in Figure 3E [25,36]. The susceptibility of proline towards ^1^O_2_ was also proven in several quenching experiments [37,38], leading to questions concerning the mechanisms behind this quenching phenomenon. The initial step in the ^1^O_2_ quenching of proline, a secondary amine, is controlled by the ionization potential (IP; capability of providing an electron). Since proline as a secondary amine has a low IP and is sufficiently soluble in aqueous environment, it can easily form charge–transfer complexes, after which either a physical or chemical decay pathway follows [38]. The physical quenching is based on the return of proline to its singlet state, while the chemical pathway involves the formation of radical species. The latter can finally result in the fragmentation of the peptide backbone adjacent to the proline residue as was stated by Schuessler et al. in 1984 [39]. However, complete mechanistic insight into the reactivity of ^1^O_2_ towards proline is still lacking.

In a biological context, amino acids are mostly present in a peptide backbone or in a protein structure, which will on the one hand influence the ^1^O_2_ reaction rate of the amino acids, but on the other hand renders the consequences of photo-oxidation quite severe. In 1903, Von Tappeiner stated that ^1^O_2_-mediated photo-oxidation of enzymes led to a loss of enzymatic or functional activity [40]. Further studies have elaborated on this topic, and it was proven that the formation of peroxides in the protein structure of, for example glyceraldehyde-3-phosphate dehydrogenase, followed by radical and non-radical mechanisms, led to the final enzyme inactivation [41]. Additionally, other consequences of protein photo-oxidation were examined. Side chain product formation, such as the presence of N-formylkynurenine (photo-oxidation product of tryptophan), could cause a change in the secondary or tertiary structure of the protein, which is one of the mechanisms that ultimately leads to a functional loss of the protein [42]. However, oxidized forms of the amino acids are not the only protein-related species resulting from ^1^O_2_ damage. For example, the heme group of catalases is highly susceptible to ^1^O_2_ (hydroxylation of the heme group [43]), resulting in different catalase conformers with more acidic isoelectric points [44]. In 1970, T. Gomyo proved that photo-oxidation of tryptophan residues in a lysosome gave rise to backbone fragmentation after one hour of illumination, illustrating another consequence of the damage of ^1^O_2_ on a protein level [26,45]. On the other hand, instead of backbone fragmentation, protein aggregates could be formed upon photo-oxidation [26]. Recently, E. F. Marques et al. proved, using mass spectrometry, that 2′ oxo-histidine can react with Trp, Lys or other His residues, leading to cross-link formation and aggregation in lysozyme. Additionally, it was shown that oxidized forms of tryptophan could cross-link to histidine or to other tryptophan moieties [46]. Additionally, S. Jiang et al. described a novel disulphide CL pathway, based on the photo-oxidation of disulphide bonds, which leads to reactive intermediates that on their turn give rise to new protein–protein cross-links (*vide infra* Section 3.1.2) [47]. To conclude, the ^1^O_2_-mediated oxidation of proteins can lead to multiple biophysical and biochemical changes, such as susceptibility to proteolytic enzymes (higher turnover), protein misfolding, changes in conformation and/or binding of cofactors, increase in hydrophobicity etc. [26], which impairs the biological function of the native biomolecules involved.

That being said, it is essential to remark that the susceptibility of amino acids towards ^1^O_2_ damage is significantly influenced by the position of the amino acid in the protein. A study by S. L Jensen et al. on five different proteins, each containing a single tryptophan at a different position in the sequence, illustrated not only the accessibility of the amino acid (solvent exposure), but also that the local microenvironment influences the reaction rate. The exact chemical environment of the susceptible amino acid within a protein can either slow down or accelerate the reaction with ^1^O_2_ either by creating a non-polar pocket that destabilizes the charged transition state or by rendering the amino acid more nucleophilic, respectively [48]. Thus, several aspects and scenarios need to be taken into account when looking at possible oxidative damage at the protein level.

### 2.2. Nucleic Acids

DNA is the key molecule carrying the genetic instructions necessary for the maintenance of all aspects of cell physiology, including development, functioning and reproduction. Alongside RNA, it plays a central role in ensuring correct synthesis of the proteins involved in the structural functions of each cell [49]. As a result, errors occurring within genomic DNA, such as DNA damage and mutations, are at the base of the carcinogenesis process [50,51,52,53,54]. It is not a surprise that extensive research has been carried out to understand the causes for potential damage to these macromolecules. The most frequently occurring damage results from DNA depurination (up to 10,000 events per cell per day in humans), involving: the hydrolysis of the glycosidic bond and results in nucleobase loss with the formation of an apurinic site [55]; single-strand breaks (occurring with a frequency of several tens of thousands per cell per day), leading to a discontinuity of one of two strands of the double helix as a consequence of the loss of a nucleotide [56]; DNA methylation, caused by external alkylating agents, with special reference to the formation of 6-O-methyl-guanine, a mutagenic nucleobase that causes the formation of a mispaired couple of bases with thymine (rather than cytosine), during the replication circles [57]. Finally, and more importantly in view of the topic reviewed here, oxidative damage can occur, with the formation of, most commonly, 8-oxo-2′-deoxyguanosine (8-oxo-dG), linked to the exposure to oxidative stress conditions. Among the over one hundred possible oxidative lesions that can occur to DNA, 8-oxo-dG is without doubt the most extensively studied and documented. It is however important to remind the reader about the existence of additional oxidation products involving DNA nucleobases, such as 2,6-diamino-4-hydroxy-5-formamidopyrimidine (also resulting from the oxidation of guanine), 7,8-dihydro-8-oxo-adenine, 4,6-diamino-5-formamidopyrimidine and ethenoadenine (resulting from the oxidation of adenine), and thymine glycol, 5-hydroxycytosine and dihydrouracil (all resulting from the degradation of the respective pyrimidine nucleobases) [58,59].

The formation of 8-oxo-dG as oxidative DNA lesion mostly occurs through a type II photosensitization mechanism. The higher sensitivity of guanine towards oxidation is due to the lower redox potential which, in contrast to the other nucleobases, makes it more suitable to undergo oxidation reactions in presence of ROS [58]. At a steady-state, the number of 8-oxo-dG lesions is esteemed in the order of 1000 per cell per day, making it one of the most common and consequently best-characterized DNA lesions [59]. It was demonstrated that ^1^O_2_ is the main reactive species responsible for the formation of 8-oxo-dG in DNA, as it seems to be reacting exclusively with the guanine rather than the other bases. The proposed mechanism involves the formation of an intermediate endoperoxide, through the Diels–Alder-like [4 + 2] addition of oxygen to the five-membered ring of the guanine. The unstable species undergoes a rearrangement, with the formation of an intermediate peroxide species, which is finally reduced to 8-oxo-dG (Figure 4) [60].

It was shown that UV-A-mediated photo-generation of ^1^O_2,_ generated in the presence of endogenous PSs, such as flavins and certain porphyrins, are responsible for the formation of 8-oxo-dG in cells through a type II photo-oxidation mechanism. Interestingly, it was more recently proposed that the DNA itself can act as a UV-A PS. This intrinsic photosensitization mechanism can additionally explain the high number of 8-oxo-dG lesions occurring per day [61].

The main issue with the formation of this oxidized guanine species is related to its structural similarities with thymine. In presence of the natural guanine, its structural features are not a problem for the correct base-pair recognition, due to the preferential orientation of guanosine in *anti*-conformation. This most stable orientation naturally avoids the mispairing of the nucleobase with adenine, due to the structural features of the DNA duplex. However, when oxidized to 8-oxo-dG, both *syn* and *anti*-conformations can be found, leading to a possible mutagenic lesion due to a G:C → T:A transversion (Figure 5) [62].

Besides its possible mutagenicity, it was demonstrated that the 8-oxo-dG adduct can be further involved in reactions with nearby nucleophiles. More specifically, the increased electrophilicity of the C5 position makes 8-oxo-dG suitable for reaction with the amino groups of proteins (e.g., lysine lateral chains), leading to the formation of irreversible protein–DNA CL adducts [63]. Using histone mutants, Bai et al. demonstrated that G-oxidation can, in some cases, produce quantitative DNA–protein adducts in nucleosome-core particles [64]. Although underestimated, this type of lesion can have deleterious consequences for the normal DNA biology, as it can disrupt the normal disposition of the double helix, blocking the transcription and replication of the DNA and limiting the repair mechanisms [65].

In addition to 8-oxo-dG, the ROS-mediated oxidation of adenine (8-oxo-dA) is also possible, albeit due to the lower oxidation potential, the estimated quantity of 8-oxo-dA formed to oxidation damage is 1/10 as compared as 8-oxo-dG [66,67]. For its minor role in oxidation-mediated DNA lesion, the 8-oxo-dA-mediated damage was overshadowed by the more prominent 8-oxo-dG-based damage. However, in a recent paper by Lee and co-workers, it was shown that the oxidation of adenine can produce interstrand cross-linking (ICL) adducts upon reaction of the oxo-dA C2 position with the N2 of nearby guanines and N3 of nearby adenines, coming from the opposite, hybridized strand [68].

### 2.3. Carbohydrates

Carbohydrates are a class of molecules, including sugars and starches, made up by simple sugar building blocks, held together through glycosidic bonds. Their main natural role consists of providing energy, as ATP molecules, through the biochemical process of oxidative phosphorylation [69]. Additionally, carbohydrates have an important role in the structural physiology of cells, as they are a constitutive part of the cell membranes, where they mediate cell–cell recognition, adhesion and signaling [70]. As compared to the other classes of macromolecules, the reactivity of carbohydrates towards ^1^O_2_ is less studied, and therefore less described in literature [71]. In 2004, Ki-Oh Hwang et al. stated that ^1^O_2_ reacts preferentially with lignin in the presence of cellulose, but that the generated radicals will eventually cause carbohydrate degradation [72]. On the contrary, sugar units are introduced in PSs, such as porphyrins [73] and are examined as possible chiral regulators of the stereoselectivity in the reaction of ^1^O_2_ with naphtalenes [74].

### 2.4. Lipids

Lipids are a distinct class of molecules (fats, hormones, oils etc.) that are involved in multiple cellular processes, such as energy storage, signaling and structural strength of the cellular membrane [25,75,76]. Oxidative damage to this class of molecules can result in several detrimental scenarios, from pore formation and increased cell membrane permeability, to protein modifications, and even the development of potentially pathological conditions [77]. Free radical independent reactions, to which the reactions with ^1^O_2_ belong, form one of the main triggers leading towards lipid oxidation.

Lipid hydroperoxides are the primary products formed upon ^1^O_2_ oxidation. In a next step, a range of cascade reactions follow that eventually result in the formation of stable oxidation products. The next paragraph will illustrate some of these cascade reactions, highlighting also their biological effects. A more elaborate mechanistic overview of the oxidation cascades occurring in lipids have been consistently reviewed elsewhere [25].

The cellular membrane, the chemical-physical barrier separating the intracellular environment from the outside, mainly consists of two types of lipophilic components, phospholipids (main constituent of the double layer [78]) and cholesterol (modulator of the membrane fluidity and permeability [79]). Both are susceptible to ^1^O_2_ oxidation and several oxidation products are formed and discussed below.

Cholesterol, one of the major regulators of the membrane’s physical properties (rigidity), reacts in an ene-type reaction with ^1^O_2_, resulting in several hydroperoxides, illustrated in Figure 6 [80]. Upon several cascade reactions, including the formation of alcohol, aldehyde and ketone groups, two electrophilic secosterol (seco-A and seco-B) compounds are obtained. The latter are promoting protein modification and aggregation, finally showing the role of cholesterol oxidation in cardiovascular and neurogenerative diseases [81,82]. Cholesterol hydroperoxides can also easily translocate to other membranes and specific sensor targets. Two aspects play an important role in this phenomenon: on the one hand, cholesterol hydroperoxides are more polar than the naturally occurring lipids, rendering them ideal to desorb in the extracellular or intracellular aqueous environment. On the other hand, transfer proteins, playing a role in lipid metabolism and membrane biosynthesis, were proven to be involved in the translocation of these hydroperoxides, resulting in an increase in oxidative damage and cytotoxicity [83].

Membrane phospholipid hydroperoxides on the other hand, are formed upon the addition of ^1^O_2_ to one of the double bonds in the unsaturated fatty acid chain. As a result, multiple hydroperoxide isomers can be formed, which subsequently decompose into more stable products. The latter can happen via several decomposition pathways, for example the reduction to alcohols (pathway A in Figure 7), followed by the loss of a hydroxy-substituted fatty acid. Figure 7 also illustrates an alternative decomposition pathway in which the hydroperoxide is transformed upon the presence of metal ions and other ROS species into a highly reactive peroxyl or alkoxyl radical, leading to fragmentation reactions that finally result in truncated phospholipids and short chain aldehydes [84]. Recent studies have shown that an increase in these truncated phospholipids were responsible for membrane permeabilization [85]. Another investigation proved the formation of pores by discovering the membrane expansion combined with the loss of optical contrast when using PSs in a membrane model system [86]. All these aspects highlight the cytotoxic aspect of the oxidation reactions of membrane phospholipids, proving their pathological implication in carcinogenesis and skin aging [77,87]. We remind the reader of other complete review works describing the biological consequences of lipid peroxidation [87].

On the other hand, the cytotoxic effect of phospholipid oxidation can also be used in antimicrobial photodynamic therapy [88,89] and as a therapeutic treatment of cancer [90]. However, the molecular mechanisms behind the cell toxicity generated by ^1^O_2_ oxidation of cellular membranes are still poorly understood. In antimicrobial photodynamic therapy, three main strategies for the allocation of PSs towards the microorganism (more specifically its membrane) can be applied [89]. First, the membrane can be targeted through the use of non-specific electrostatic interactions, which renders the selectivity towards bacteria far from absolute. Additionally, more specific targeting ligands for the allocation towards specific membrane proteins can be used. As a third strategy, membrane-disrupting ligands, such as well-known antibiotics (ex. Vancomycin to treat Gram-positive bacteria), can be attached to the PS [89]. A more elaborate discussion on these different types of PS ligands for antimicrobial activity can be found in Section 4.

## 3. ^1^O_2_ in Bioorganic Chemistry Applications

Due to the increasing availability of PSs capable to generate ^1^O_2_ in solution, the number of possible applications has become larger in the past decades. In this context, ^1^O_2_-based approaches have found application in the field of bioorganic chemistry, producing methodologies available for oligonucleotide and protein labelling, as well as CL and biosensing.

### 3.1. Peptide and Protein Modifications: Labelling, Cross-Linking and Knockdown

#### 3.1.1. ^1^O_2_-Based Peptide and Protein Labelling

Labelling a specific (type of) protein or peptide with a dye, radioactive handle or property enhancing moiety has gained substantial interest over the years, since it is a powerful tool to analyze the dynamics of living systems and to synthesize new protein-based biomaterials [91]. Throughout the years, multiple labelling methods have been reported, for example the azide-alkyne click reaction [92,93], the tetrazine-alkyne inverse electron demand Diels–Alder reaction [94] and labelling of native moieties in peptides and proteins, such as carbonyl labelling by hydrazines or alkoxyamines [95], chemoselective labelling of tyrosine [96,97], tryptophane [98], cysteine [99,100] and lysine [101]. A detailed overview of such biorthogonal bioconjugation approaches are given by C. S. McKay et al. [102]. Alternatively, enzymatic methodologies, based on the application of, for example sortase A [102,103], protein farnesyltransferases [104] or lipoic acid ligase [105], can be used in this context. Research towards straightforward, high yielding, synthetically simple and non-toxic labelling methods is still ongoing. As an addition to the toolbox of methods discussed above, we discuss here the use of ^1^O_2_-mediated peptide/protein labelling methods.

In 2016, E. Antonatou et al. applied the ^1^O_2_-mediated oxidation of furan, followed by the reaction with a hydrazine derivative as a new type of labelling strategy [106]. The furan is in the first step oxidized by ^1^O_2_ (see Figure 2 in the introduction for the reaction mechanism), which is generated in solution by irradiation of the PS Rose Bengal (RB). In the second step, the generated electrophilic keto–enol moiety reacts with a suitable alpha-effect nucleophile in order to form a pyridazine or pyrrolidone ring system (Figure 8A) [106]. This paper illustrates that the amount of oxidative damage is dependent on the photo-oxygenation conditions (concentration and type of PS, irradiation time) and that in any case, a careful consideration and optimization of these is required in order to keep the collateral damage to the minimum.

A careful evaluation of the photo-oxidation conditions was performed in a recent work of C. Wang et al. [108]. In 2016, they described the application of the strong PSs MB (^1^O_2_ quantum yield (QY): 0.52 in water [109]) and RB (^1^O_2_ QY: 0.76 in water [109]) as a turn-on catalyst for tetrazine ligation. More specifically, red-light irradiation of MB catalyzed the oxidation of dihydrotetrazine to tetrazine, that subsequently reacts with a strained alkene or alkyne in an inverse electron demand the Diels–Alder reaction (Figure 8B). This ligation methodology was successfully applied, as proven by confocal microscopy, for the fluorescent labelling of dihydrotetrazine-containing fibers [107]. Now stated in their recent work, further in vivo applications were hampered by the phototoxicity of MB, leading to extensive oxidative damage. This was illustrated in a proof-of-concept experiment, in which the protein thioredoxin (Trx, 12 kDa) was exposed to the MB photo-oxygenation conditions applied in their ligation protocol. As LC-MS experiments clearly showed protein damage (Figure 8C), they decided to adopt a milder PS, such as a Si-Rhodamine derivative [108]. Although, the ^1^O_2_-generation by these dyes can be more controlled, oxidative damage is still observed. Similar observations were made in a recent work by L. Miret et al. where ^1^O_2_-mediated protein–protein CL was studied using either RB or rhodamine derivatives (*vide infra*) [110].

#### 3.1.2. ^1^O_2_-Mediated Peptide and Protein Cross-Linking

Protein–protein interactions (PPI’s) are crucial in various biochemical processes, such as signal transduction, sensing the environment and keeping control of the cellular organization [111]. Research towards these PPI’s increased our knowledge on the complex cellular organization and on the development of diseases [112,113]. In several pathological conditions, PPI’s are upregulated or downregulated or abnormal PPI’s with pathogenic proteins take place [114]. The inhibition of these protein–protein interactions has high clinical significance, highlighting the need to study and mimic these types of interactions. Not only is the study of PPIs relevant from a clinical point of view, but it also helps to elucidate the function of proteins and to study their binding sites.

##### Photo-Oxidative Cross-Linking of Peptide/Protein Interactions

Chemical CL, combined with analysis by mass spectrometry or NMR, constitutes an important tool to lock and study protein–protein interactions [115,116,117]. One group of protein–protein cross-linking reactions are based on the use of irradiation to activate a photoactivatable agent/amino acid, bearing diazirines [115,118], benzophenones, phenols or phenylazides [119,120,121]. In recent years, the use of photoactivatable groups for protein–protein CL was combined with click chemistry in order to enrich for the cross-linked complex in mass analysis [122,123,124,125,126] or to introduce a fluorescent tag onto the complex [127]. In the context of protein–protein CL, methodologies relying on the photo-induced production of ^1^O_2_ have also been reported for peptide and protein modification.

A first type of methodology for protein CL, recently described by S. Jiang et al., relies on the formation of new intermolecular disulphide bridges in the presence of ^1^O_2_, upon reaction with disulfides in presence of a free thiol (coming either from a free protein or glutathione) [47,128]. A possible CL mechanism is illustrated in Figure 9A. This strategy was applied in several proof-of-concept experiments with different disulphide-containing proteins (C-reactive protein, α-lactalbumin, lysozyme and β-2-microglobulin) and the free thiol containing GAPDH to prove the capability of ^1^O_2_ to cross-link proteins that did not contain a free thiol at first. S. Jiang et al. illustrated the loss of activity of the GAPDH protein upon cross-link formation. This new protein CL method could then further help to explain the mechanisms behind the accumulation of cross-linked protein species in various pathological conditions. However, applying this method for bioconjugation is more complex due to the lack of specificity in case of the presence of multiple disulphide bridges in the target, the influence on the structure of the target upon disruption of the original disulphide bridge and the oxidative damage caused by the use of PS in solution.

Next to the S-glutathionylation, explained above, it was investigated whether glutathione can also be covalently conjugated to oxidized tyrosine residues in peptides and proteins after being exposed to ^1^O_2_. P. Nagy et al. suggested a mechanism in which tyrosine hydroperoxides, formed upon ^1^O_2_ addition, can form bicyclic compounds with (N-terminal) amine functionalities, resulting in the formation of an α,βunsaturated carbonyl functionality. Subsequently thiol-containing substrates, such as glutathione, will react via an 1,4 Michael addition onto the double bond (Figure 9B). However, it was found that the cross-link formation was reversible, and further investigations for the application of this methodology in a cellular context are required [129]. Tyrosine can not only form a cross-linked structure with thiol-containing compounds, but also dityrosine adducts are generated in the presence of ^1^O_2_ [130], although earlier work states the formation of dityrosine cross-links only through radical reactions and type 1 photosensitization [131].

Additionally, cross-link formation with other ^1^O_2_ sensitive amino acids in proteins was investigated in order to elucidate the ^1^O_2_-mediated in vivo cross-links possibly formed in various diseases. In 1978, J.R. Lepock et al. highlights the role of ^1^O_2_ in the photo-induced CL observed in membrane proteins, labelled with the fluorophore and PS fluorescein isothiocyanate [132]. A few years later, J. D. Goosey et al. demonstrated the role of ^1^O_2_ in the non-disulphide CL of lens crystallin polypeptides, seen during aging and cataractogenesis [133]. However, the specific structure of these cross-links was not clarified. In 1996, preliminary studies on two model systems, histidine- and lysine-containing N-(2-hydroxypropyl)methacrylamide copolymers and ribonuclease A, suggested the involvement of His and Lys in photodynamic CL of proteins [134,135]. In 2014, Liu et al. characterized the histidine–histidine cross-link adduct, formed upon photo-oxidation of IgG1 antibody [136]. Other types of cross-links, involving the amino acids trypthophan, tyrosine, lysine and histidine, in proteins exposed to photo-oxidation were elucidated by Mariotti et al. [130]. Recently, E. F. Marques et al. proposed a detailed reaction mechanism on the formation of various types of non-disulphide intermolecular cross-links upon the exposition of lysozyme towards RB, oxygen and light (Figure 9C). It was proven that ^1^O_2_ was involved in the generation of cross-links between oxidized histidine, histidine, lysine and kynurenine derivative of trypthophan. Additionally, cross-links between oxidized trypthophan and the open ring kynurenine derivative were seen [46]. A small overview on the types of protein cross-links and high molecular mass aggregates is given by E. Fuentes-Lemus et al., combined with the relation of photo-oxidation of protein with pathological conditions [137].

Within our research group, the photochemical furan-based oxidation technology described by K. Hoogewijs et al. [138], has been applied for the ligation [106] and cyclisation [139] of peptides as well as the covalent trapping of protein–protein interactions [110]. In a peptide ligation strategy, furan was attached onto the C-terminus of one peptide and successfully ligated with a hydrazide containing peptide in the presence of light and RB as PS. The general principle is illustrated in Figure 9D. Moreover, in this case, an important note on the oxidative damage of other amino acids needs to be made, since it was shown by E. Antonatou et al. that cysteine and methionine in close proximity of the furan were also oxidized in the presence of ^1^O_2_ [106]. Collateral damage was also shown in case of the use of the furan oxidation applied in the context of weak protein–protein interactions. However, the use of the softer PS Rhodamine B resulted in less oxidative damage and kept the polymerization function of actin, which was one of the cross-linking partners [110], thus highlighting the importance of the exact reaction conditions. Although the oxidative damage depends on the irradiation conditions (^1^O_2_-QY of the PS and its concentration, irradiation time and type of light source used), it was found that furan is more sensitive towards ^1^O_2_ compared to any amino acids [140,141]. Recently, A. Manicardi et al. reported on a related methyl-furan-based proximity-induced peptide ligation method that does not require the use of ^1^O_2_ and thus avoids oxidative damage [142].

Protein–protein cross-link formation, mediated by ^1^O_2_ production, also has clinical relevance as shown by its application in the treatment of keratoconus. Keratoconus is a congenital eye-disease in which the stiffness of the cornea decreases over time, leading to a more bulged-out shape of the cornea. This results in a distorted vision due to incorrect light focusing [143]. In the past 20 years, corneal CL has been applied to increase the rigidity of the corneal stroma and thus to halt further progress of the disease. The treatment consists of the use of riboflavin as PS and ultraviolet-A light for the generation of ^1^O_2_. Subsequently, the latter reacts with collagen fibers in the stroma and proteoglycans in the extracellular matrix [144,145]. Investigation of the corneal CL mechanisms by A. S. McCall et al. proves the role of ^1^O_2_ and carbonyl-moieties in the stroma and suggests the involvement of histidine, hydroxyproline, hydroxylysine, tyrosine and threonine moieties [146]. However, ^1^O_2_ is not only involved in the formation of corneal cross-links, but it is also responsible for keratocyte apoptotic cell death, an adverse effect of this treatment [147]. Although this seems a drawback, the lower inflammatory response mediated by an apoptotic rather than a necrotic cell death limits the biotoxicity of ^1^O_2_ in this context [148]. Additionally, it was proven that although the cell density of the keratocytes reduced after the treatment, it returned to standard values within a year [149].

##### Endogenous ROS-Mediated Ligand-Receptor CL

In the past few years, it has become clear that ^1^O_2_ and the umbrella class of reactive oxygen species (ROS) are involved in various levels of the cellular (mal-)functioning. ROS inhibit or activate proteins, are involved in the promotion and suppression of inflammation and play an important role in immunity by eliminating microorganisms [150,151]. Additionally, increased levels of ROS are present in almost all cancer types involved in aspects of tumor development and growth [152]. ROS are endogenously produced by several sources, such as the different isoforms of the NADPH oxidases (NOX enzymes), the mitochondrial respiratory chain and lipo- and cyclo-oxygenases [150].

Such endogenous increased levels of ROS species, expressed by NOX enzymes on cancer cells, can also be used for CL of ligands onto cell-surface receptors [153]. In 2017, our group illustrated that ROS activity can be used to endogenously oxidize furan moieties within a peptide, leading to keto–enol formation and generation of CL species (Figure 10). The role of the NOX enzymes in this context was proposed and confirmed by the lower cross-link yield observed when using different NOX-inhibitors in two cell model systems [153]. However, more research towards the mechanisms behind this ROS activation strategy and the role of NOX is required.

#### 3.1.3. Protein Knockdown

In order to clarify functions of endogenous cellular proteins, techniques for selective inactivation of specific protein functions have been developed. One example worth mentioning in the context of this review is chromophore/fluorophore-assisted laser inactivation also referred to as CALI. In this method, a small dye is conjugated to an antibody while the protein of interest or a fluorescent protein fused to the target protein is introduced or expressed in the cell. Upon light irradiation of the area of interest, ROS, including ^1^O_2_, are produced which in turn inactivate the proximate proteins. Since the free radical species are short-lived, only proteins immediately adjacent to the irradiated chromophore are damaged. This technique thus provides a spatially and temporally controlled loss-of-function tool for cell and developmental biology [154,155]. Later, fluorophores were also used, and the technique is then referred to as FALI [156].

Originally, CALI was developed with the dye Malachite green as a chromophore. This dye was conjugated to a target-specific non-function-blocking antibody and this construct was subsequently microinjected into the cell or bound to cell-surface determinants [157]. In the following years, dyes that are better ROS generators, usually xanthene-based, were explored as well. An example is given by the labelling of antibodies with fluorescein-isothiocyanate to inactivate β-galactosidase and the microtubule motor kinesin in vitro [158].

In 2008, T. Nagano et al. presented a design strategy for PSs with an environment-sensitive off/on switch for ^1^O_2_ generation, to improve the specificity of protein photo-inactivation. In its unbound state, ^1^O_2_ generation is quenched by photo-induced electron transfer. Only in the hydrophobic environment that is provided by the target protein, ^1^O_2_ can be generated. This was demonstrated via the inactivation of the inositol 1,4,5-triphosphate receptor using an environment-sensitive PS-conjugated inositol 1,4,5-triphosphate receptor ligand. [159].

Over the last few decades, new techniques were developed to add functionalities (such as fluorescent dyes and photosensitizers) to proteins in vitro as well as in vivo. Some prominent examples are the Halo-Tag, biarsenical-tetracysteine system, the β-galactosidase-, CLIP- or SNAP-tag labelling approaches [160]. In the context of CALI, the biarsenical-tetracystein labelling system allowed specific binding of FlAsH-EDT2 and made inactivation of Synaptotagmin I and connexin-43 possible [161,162]. Moreover, inactivation of α- and γ-Tubulin was done successfully in living cells, via a SNAP-Tag fusion protein that was linked via a thioether bond to an O^6^-benzylguanine bearing a fluorescent dye at the paraposition (See Figure 11) [163]. Upon light irradiation, singlet oxygen was generated, which resulted in the inactivation of those proteins. Protein inactivation mediated by fluorescent proteins such as enhanced green fluorescent protein and mini singlet oxygen generator (MiniSOG) have been described as well [164,165].

### 3.2. Oligonucleotide Modifications: DNA and RNA Labelling, Cross-Linking, and Targeting

#### 3.2.1. Oligonucleotide Labelling Methodologies Featuring ^1^O_2_

Detection, imaging and purification of nucleic acids are fundamental processes for understanding the role and mechanism of targeted sequences in biological processes and for the correct diagnosis of diseases depending on an altered genetic sequence. Labelling of the sequence of interest is often required to make these processes possible. Labelling of the desired oligonucleotide can be done through the incorporation of a radionuclide (typically 32P and 35P, incorporated in the phosphate backbone of the sequence) [166]. Alternatively, chemical modifications on the 5′/3′ termini of the sequence can be introduced, either synthetically or enzymatically. Common modifications include the introduction of fluorophore tags to enable the imaging of the sequence in a tissue or in cellulo [167]; the introduction of a biotin tag, which enables the recognition with avidin proteins, that can be used either for detection (in the case of ELISA-like assays) and target pull-down purposes (avidin/streptavidin beads) [168]. Albeit limited, methodologies featuring the production of ^1^O_2_ as a mediator to introduce modifications on an oligonucleotide, are also reported. An example of this strategy is given by the introduction of an azide moiety through a β-difluoroalkylamine moiety, which enables post-labelling RNA biotinylation through click chemistry. The proximity-induced introduction of the azide tag is made possible through the ^1^O_2_-mediated photo-oxidation of a guanine residue of the RNA strand, which is able to react with the amino function of the azido-β-difluoroalkylamine probe [169].

#### 3.2.2. Oligonucleotide Cross-Linking and Targeting Methodologies

The use of bifunctional electrophilic compounds, such as nitrogen mustards, finds a well-established application in chemotherapeutic drugs able to induce a covalent cross-link between nucleophilic residues of the DNA. These approaches, however, lack in selectivity, as they lead to the random formation of covalent bonds between two nucleophilic moieties, ultimately leading to a generalized toxicity and severe side-effects. The use of pro-reactive moieties can help to overcome the selectivity-related problems, and in order to direct the cross-linker probe to the desired target, these moieties can be effectively attached to an oligonucleotide or analogue, exploiting the base-pair recognition. Besides for chemotherapy, the formation of selective ICL species can be exploited for the design of capture probes, leading to other important applications such as the design of pull-down probes to fish out target DNA/RNA strands from biological samples or for biosensing purposes. In this context, light-triggered methodologies exploiting the in situ formation of ^1^O_2_ to activate pro-reactive species are available. The possibility to modulate the wavelength for generating the active oxygen and the use of harmless visible light, make its use advantageous as compared to other available triggered methodologies, often requiring use of harmful UV light, or to the trigger-free methodologies relying on highly reactive moieties.

Back in 2009, our group reported on the incorporation of a furan moiety into an oligonucleotide strand, able to induce selective ICL upon its activation by N-Bromosuccinimide [170]. The species formed is a highly reactive electrophilic keto–enol, able to react with the nucleophilic residues of the DNA nucleobases adenine, cytosine and guanine [171]. Given the spatial orientation of the exocyclic amines when forming the double helix, the ICL showed higher yields when the furan-modified nucleobase of the probe is facing a cytosine and adenine. In a later work, the insertion of the furan moiety in 2′ position of a uridine lead to the discovery of C-selective ICL probes [170]. In a follow-up of this study, furan activation was then achieved by ^1^O_2_-mediated photo-oxidation, in presence of MB as PS [172], and later on extended from DNA to RNA [173]. The technology was further successfully applied in a DNA-templated fashion, upon insertion of the PS in another DNA strand, adjacently hybridized to the target [174,175] (Figure 12B). A similar concept of exploiting DNA as template, was applied through the co-localization of MB, a weak G-Quadruplex (G4) binder, with a furan-containing G4-ligand, in order to enable selective alkylation of G4-DNA over dsDNA, triggered by red light [176]. Beside the original purpose of ICL and DNA/RNA alkylation, furan technology was also exploited for the covalent immobilization of oligonucleotides on surface, as demonstrated by SPR measurements [177]. This chemistry was at the same time extended towards unnatural oligonucleotide derivatives, such as PNAs, for the realization of PNA-DNA (Figure 12A) CL systems, using NBS to trigger the furan oxidation, and further adopted by the Vilaivan group for the realization of acpc (pyrrolidinyl)-PNA probes able to cross-link to a DNA target [178]. Finally, we exploited furan chemistry for the realization of an oligonucleotide-templated PNA–PNA ligation reaction, which can be conveniently used for the realization of constructs on surface as well as for the detection of short oligonucleotides [179].

In another application in this context, Summerer and colleagues took advantage of the genetic encoding of a furan moiety into tRNA^Pyl^ and Pyrrolysyl-tRNA synthetase, to enable red-light-induced ^1^O_2_-mediated protein–RNA cross-linking, using MB as external PS [180] (Figure 12C). In a more clinical perspective, furan methodology has found a recent application in tumor-targeting, showing that therapeutic applications come within reach. Shi and co-workers recently designed cross-linking probes to target cytoplasmic RNA, exploiting the functionalization of a cyclic-arginine-glycine-aspartic acid (RGD) peptide, able to bind α_v_β_3_ integrin and get internalized in the tumoral cells, with a furan side chain. The additional functionalization of the peptide with a cyanine reporter could allow the imaging of tumors in vivo. Here, photo-activation was ensured by the co-administration of MB. Although it is not clear whether the therapeutic effects are linked to selective cross-linking of the peptide to the cytoplasmic RNA, or a generalized alkylation of the targeted cells, the methodology proved to be effective in inducing tumor suppression in murine model [181].

Beside furan technology, other applications in DNA cross-linking relying on ^1^O_2_, are reported [182]. An example is given by the chemistry developed in the Greenberg lab, where the incorporation of phenyl-selenide derivatives into synthetic oligonucleotides was exploited for selectively forming an ICL with an opposing adenine-containing DNA target strand [183,184]. ^1^O_2_ production, induced upon irradiation in the presence of RB, causes the oxidation of selenide to selenoxide, which further undergoes a [2,3] sigmatropic rearrangement, leading to the formation of an electrophilic methide, able to react with the nucleobase and leading to the formation of an ICL species. Yavin and colleagues proposed an alternative approach to the strategies illustrated above. Taking advantage of the high sensitivity of guanine towards ^1^O_2_-mediated photo-oxidation, they synthesized peptide nucleic acids (PNAs) bearing a strong PS (RB) on a complementary G-rich DNA target [185]. The group illustrated the possibility to selectively induce oxidation of the target strand, with the subsequent formation of ICL products, presumably due to the reaction of 8-oxo-dG with the nucleophilic lysine introduced on the PNA strand (as described above in Section 2.2) [63,186].

### 3.3. ROS Cleavable Linkers in Drug Delivery and Prodrug Applications

Though ^1^O_2_ is widely used in the context of photodynamic therapy, the damage that it induces in tumors is still both temporarily and spatially limited. This is because ^1^O_2_ cannot diffuse beyond the cell diameter. For that reason, it is possible that in a heterogeneous tumor, cells that escaped ^1^O_2_ damage can regrow after PDT treatment. In order to avoid the latter, prodrug concepts were developed. This type of prodrug is composed of a PS, anti-cancer drug and an ^1^O_2_ cleavable linker. By illuminating the prodrug, ^1^O_2_ is generated, which results on the one hand in the damage of the tumor and on the other hand in the release of the anti-cancer drug. Subsequently, the released drug can lead to spatially broader and temporally sustained damage so that surviving cancer cells after PDT are killed. In this section, we will not give an overview of all developed prodrugs. Instead, we will focus more on the principle and the type of linkers suitable for tese kind of applications [187].

Electron rich heteroatom substituted olefines have been widely explored in the context of ROS cleavable prodrugs. Examples thereof are vinyl dithioethers, vinyl diethers and vinyl monoethers. With these olefin-type linkers, drug release can be obtained via irradiation of the prodrug with low energy light which induces a 1,2-cycloaddition reaction. Multiple studies that investigate the kinetics and irradiation conditions needed for these compounds have been published [188,189,190]. However, some disadvantages of these type of linkers should be taken into account. For example, synthesis routes for vinyl dithioethers and vinyl diethers are very scarce and the required reaction conditions result in low yield and non-stereospecificity [191]. Therefore, Y. You et al. developed a click and photo-unclick chemistry of amino-acrylates for low energy-controlled release of active compounds [192,193,194,195]. In addition, it was demonstrated that it is possible to damage the surviving cancer cells over and beyond the spatial and temporal limits of ^1^O_2_ [196]. Another strategy was reported by C. F. Barbas III et al. and is based on ^1^O_2_-mediated transformation of arylthiolanes into aryl ketones [187].

Additionally, in the design of photo-triggerable drug vehicles, the aforementioned ROS cleavable linkers are used. W. Jong Kim et al. published a nice example thereof. They synthesized a biocompatible amphiphilic block copolymer micelle that bears a vinyldithioether linker at the core-shell junction. Upon light irradiation, the polymeric micelles that were co-loaded with a PS disassemble, resulting in the release of the anti-cancer drug [197]. Recently, a photolabile spherical nucleic acid for carrier-free and near-infrared photo-controlled self-delivery of small interfering RNA (siRNA) and antisense oligonucleotides (ASOs) was also published. By incorporating a thioketal linker between the SiRNA and the peptide nucleic acid-based ASO, on-demand disassembly of photolabile spherical nucleic acid in tumors cells is possible once the ^1^O_2_ is produced by the inner PS upon NIR irradiation [198]. Furthermore, nanoparticle type vehicles also have been functionalized with a ROS cleavable linker in order to obtain light-triggered drug release (see Figure 13) [199,200,201,202,203].

### 3.4. Biosensing Applications Featuring ^1^O_2_ Generation

Biosensors are devices designed to convert a molecular recognition event into a detectable signal. Their biomedical importance is related to the need of detecting certain macromolecules in a sample, whose presence is related with the onset and development of pathologies [204]. Among the possible detection methods, an important role is played by electrochemical biosensors, which translate the biomolecular event into an electric signal [205]; fluorescence-based biosensors, which enable detection due to a difference of the fluorescence intensity in the presence of the target molecule [206]; colorimetric sensors, which provide a low-cost read-out that can be detected with bare eyes, being based on a mere color change in the presence of the target [207]; or detectors based on chemiluminescence, which exploit the light emission from electronically excited mediators returning back to their ground state [208].

Concerning detection strategies, the generation of ^1^O_2_ has recently been exploited for the realization of sensors, able to spot the presence of target oligonucleotide sequences, proteins, small molecules and even whole cells in a sample. The use of ^1^O_2_ proved to be versatile in this context, as it can be used as a direct read-out species *per se* or exploited for the oxidation of reported molecules used as redox indicators for electrochemical biosensors. An example of a ^1^O_2_-based device is described by Zhang and colleagues, who described the sensitive detection of the BRCA1 gene down to attomolar level (10 aM) (Figure 14A) [209]. In this work, a gold electrode is functionalized with a capture DNA probe, complementary to BRCA1. When the target is present in the sample, it can form a stable duplex with the capture probe. The addition of a DNA-intercalating PS (e.g., Ethidium Bromide) and the subsequent ^1^O_2_ production upon light irradiation, cause the cleavage of the DNA strands and a lower electrode blockade, as a consequence of the increase of the [Fe(CN)_6_]^3−/4−^ redox current (used as a redox indicator in cyclic voltammetry). The same group used a similar approach for the detection of lysozyme protein: in this case the electrode was functionalized with a DNA complementary to the lysozyme aptamer. In the presence of the enzyme, the aptamer is sequestered from the dsDNA complex, resulting in a higher blockade of the electrode (Figure 14B) [210]. In a relatable manner, molecular beacon structures were employed for the detection of target DNA through cyclic voltammetry, using dopamine as a reporter. The capture probe in this case consists of a molecular beacon, functionalized with MB as PS and attached onto a gold nanoparticle. In absence of the target, the molecular beacon conformation causes a fluorescence quenching, with lower ^1^O_2_ production. In contrast, upon addition of the target, the fluorescence emission of MB is restored, and the production of ^1^O_2_ under continuous light irradiation causes the oxidation of the reporter, absorbed onto the gold surface (Figure 14C) [211]. The de Wael group reported another elegant example of a DNA electrochemical sensor based on ^1^O_2_ generation. In their work, the group exploits commercially available and known DNA probes bearing a chromophore (type II PSs) and explores the detectability of these conjugates onto a gold electrode in different set-ups: probes directly captured on the gold surface, dissolved in solution and captured on a surface of magnetic beads, functionalized with complementary capture probes. Here, hydroquinone was used as a redox marker, allowing amperometric detection of the target [212].

The detection of nucleic acid targets exploiting the generation of ^1^O_2_ was also achieved through colorimetric methodologies. Dong et al. reported on a biosensor exploiting the ^1^O_2_-mediated oxidation of 3,3′,5,5′-tetramethylbenzidine (TMB), to give a fast read-out upon color development in the presence of the target DNA. The PCR-amplified DNA samples were irradiated in the presence of the DNA-intercalating PS SYBR green-I (SG), using TMB as chromogenic substrate. This allowed them to reach a low 1aM LOD, which is comparable with the results obtained by qPCR technique [213]. In a similar way, the formation of a dsDNA–SG complex was used for the detection of DNA down to 1.5 pM, exploiting ferricyanide as mediator to allow the oxidation of luminol, with a subsequent increase in chemiluminescence as read-out [214] (Figure 15A).

Protein biosensing through ^1^O_2_ production has also found recent application in literature. A first example relies on the photosensitized ^1^O_2_ to induce chemiluminescence in the presence of thrombin. In this work, Jiang et al. presented an aptamer-based sensor, by trapping a phthalocyanine dye (PS) onto silica nanoparticles. In their ELISA-like assay, the ^1^O_2_ generated in the presence of the PS could react with a methyl Cypridina luciferin analog, resulting in an increased chemiluminescence [215].

Other applications featuring the partnership between oxygen’s excited state and chemiluminescence are reported for the detection of EGR1 Zinc-finger protein. Here, Zn2+ is sequestered from the protein domain by a hollow protoporphyrin IX (PPIX) and the resulting ZnPPIX (immobilized between laptonite nanosheets), enabled the electrochemiluminescence (ECL) detection of the analyte with a low pM detection limit (0.48 pM) (Figure 15B) [216]. For the detection of carcinoembryonic antigen (CEA), a sandwich ELISA-like assay was developed, exploiting the FRET emission of PS-containing polystyrene nanoparticles (functionalized with streptavidin), and quantum dot-doped nanoparticles (decorated with an anti-CEA monoclonal antibody). In the presence of the analyte, the FRET emission allowed the light-induced ^1^O_2_ production, which triggers chemiluminescent emission from the quantum dots [217]. Other very recent applications in this context were developed by the Zhang group, who realized ECL-based biosensors for the detection of polynucleotide kinase (PNK), an enzyme implicated in the oligonucleotide metabolism [218]. In a first work, they functionalized an Au-carbon glass detector with DNA to allow deposition of porphyrin-containing metal-organic nanoparticles (NMOFs), decorated with complementary DNA. The assay is based on a competitive ECL mechanism, between the luminescence generated by ^1^O_2_ (produced by the porphyrin-containing NMOFs) and the oxidation of luminol by the presence of ROS. In presence of the PNK binding the DNA, the NMOFs is gradually detached from the surface, so that the ratio between the ECL values change, allowing quantification of the enzyme activity (Figure 15C) [219]. The same principle was used in a more recent work from the same group, in which PNK activity was assessed through the competitive reaction between luminol and 2D copper-based, porphyrin-containing NMOFs nanosheets [220].

**Figure 15 molecules-27-00778-f015:**
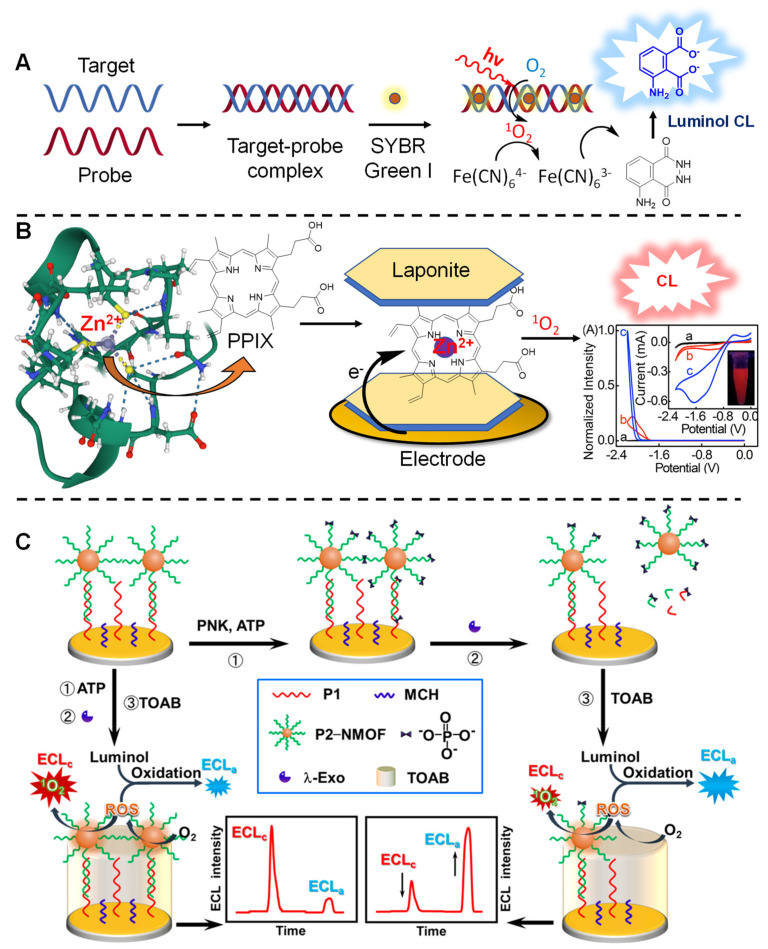
Electrochemiluminescence-based detection methodologies, featuring ^1^O_2_ generation. (**A**) Detection of a complementary DNA target based on CL given by oxidation of luminol, proposed in ref. [214]. (**B**) Detection of Zn-finger proteins showed in ref. [216]. (**C**) Competitive ECL-based detection of PNK described in ref. [219]. Figure adapted with permission from [219]. Copyright 2022. American Chemical Society.

Besides for macromolecules, where dysregulated presence and activity is often regarded as biomarker for human-related pathologies, efforts have been spent for the realization of biosensors to detect the presence of small molecules, such as toxic byproduct contaminating diaries and industrial products, or metabolites and components of human organism. A sandwich ELISA-like detection platform was developed by Guo and colleagues for the detection of bisphenol A (Figure 16A). The assay was again based on the ^1^O_2_-mediated chemiluminescence, possible only due to the proximity effect resulting from the recognition of the analyte, that holds together a PS-containing nanoparticle (donor) and Thioxane-Eu(III) nanoparticles (acceptor) [221]. A colorimetric assay based on the oxidation of TMB in the presence of melamine (a contaminant illegally added to milk products to increase the nitrogen content), was developed by Hu et al. This elegant detection system is based on the ability of melamine (both donor and acceptor of hydrogen bonds) to hold together two T-rich sequences in a duplex, acting like a bridge between two thymine units from opposite strand. Upon addition of an intercalating PS (e.g., SYBR-Green), the TMB oxidation is enhanced only when the dsDNA is formed, i.e., only in presence of the analyte [222] (Figure 16B).

Finally, an example of an electrochemical biosensor for the detection of glutathione (GSH), was recently reported by Gao and co-workers, taking advantage of Zinc(II) tetraphenylporphyrin (ZnTCPP), deposited onto an electrode and functionalized with graphene oxide. At the same time, hydroquinone (HQ) was used as a current amplifier for the amperometric detection of GSH. The oxidation of GSH into GSSG dimer due to production of ^1^O_2_ by ZnTCPP, goes along with the oxidation of HQ into a benzoquinone (BQ). The successive reduction of the BQ into HQ leads to the final amplification of the current, closing the redox cycle [223].

## 4. How to Tame the Bullet

As discussed in Section 2, ^1^O_2_ is a ROS species able to induce substantial damage to several biomolecules. Hence, its use could be limited by the possible collateral damage, particularly when the integrity of the payload plays an important role. As a result, methodologies capable of tailoring ^1^O_2_ production to the minimum amount required have been developed in order to minimize the oxidative damage, which in a biological context can be key for translating the application to the clinic. In this section, we discuss those methodologies that feature the confined production of ^1^O_2_, exploiting the use of genetically encoded PSs or the incorporation of a PS into a DNA (or analogue) sequence, a peptide or protein in order to limit the oxidative damage to the target.

### 4.1. Genetically Encoded PSs

Currently, genetically encoded PSs can be categorized into two groups. The first group contains fluorescent proteins, which exhibit a green fluorescent protein (GFP) like structure, whereas the second group encompasses flavin-binding fluorescent proteins derived from the light-oxygen-voltage (LOV) photoreceptor domain [224]. Since this type of PSs are encoded by a gene, they can be expressed in a spatially and temporally regulated manner, under a promotor of choice, and fused with the desired protein of interest or localization signal. In addition, they are not reactive towards cellular components and they do not interfere with any cellular pathways in commonly used prokaryotic or eukaryotic model systems.

KillerRed and the ^1^O_2_ Generator (miniSOG) were the first described genetically encoded PSs. KillerRed was reported in 2006 by Bulina et al. [225]. It was derived from the hydrozoan chromoprotein anm2CP and is a dimeric red fluorescent protein with fluorescence excitation/emission maxima at 585/610 nm. Under a chromophore-assisted laser or light inactivation (CALI), KillerRed was originally used because of its believed high efficiency in generating ROS [155,226]. However, it is now acknowledged that it does not work through ^1^O_2_ as its ^1^O_2_ quantum yield is negligible [227].

MiniSOG was introduced by Shu and co-workers [228]. MiniSOG is composed of 106 amino acids. It was engineered from the LOV 2 domain of Arabidopsis phototropin 2 and is thus able to noncovalently bind to flavin mononucleotide. Consequently, miniSOG can absorb blue light, fluoresces green light and its excitation results in the generation of ^1^O_2_. Originally, it was developed to catalyze the polymerization of diaminobenzidine into an osmiophilic reaction product resolvable by electron microscopy. In 2013, the phototoxicity of miniSOG in cancer cells was assessed for the first time by Ryumina and co-workers [229]. They were able to demonstrate that miniSOG is an excellent genetically encoded PS for mammalian cells in vitro. In the same year, inhibition of synapses with CALI proved that miniSOG is also a helpful tool in neuroscience [230]. Since then, the use of miniSOG has gained more interest and several new applications were reported [231,232,233,234]. Recently, A. Care et al. reported a first-of-its-kind encapsulin nanoreactor that is able to house miniSog and that upon blue-light irradiation is able to produce ^1^O_2_ which induces a phototoxic effect on lung cancer cells. Additionally, they also observed an additive effect between the encapsulin nanoreactors and the miniSOG, which provides a unique advantage over the use of free miniSOG alone [235]. Moreover, in the context of drug delivery applications, these encapsulin nanoreactors loaded with miniSOG show to be promising [236].

### 4.2. PS Conjugates

#### 4.2.1. Peptide Conjugates

Within the group of peptide–PS conjugates, three categories can be distinguished. The first one makes use of antimicrobial peptides, the second one is based on cell penetrating peptides (CPP) and finally the last one encompasses peptides that are receptors for growth factors.

Antimicrobial peptides (AMPs) usually include two or more positively charged residues provided by arginine, lysine or histidine (in an acidic environment). These positively charged peptide–PS conjugates are used to increase the affinity of the PS for the surface of Gram-negative bacteria (see Figure 17). In comparison to Gram-positive bacteria, Gram-negative bacteria have a more complex cell envelope, resulting in a permeability barrier that restricts the binding and penetration of many PSs [237].

In 1998, Tayyaba Hassan et al. hypothesized that conjugating positively charged poly-L-Lysine (20 residues) to the PS chlorin e6 (ce6^2^) would result in selective photo-destruction of both Gram-positive and Gram-negative bacteria while the host’s epithelial cells would be spared [238]. In their study, they compared the uptake and phototoxicity of polycationic (pL-ce6), neutral (pL-ce6) and polyanionic (pL-e6) conjugates to unconjugated ce6, ce6-monoethylenediamine monoamide and a mixture of pL and e6 in an oral epithelial cell line. Their results suggested that an increased cellular penetration of the PS through the outer membrane of Gram-negative bacteria was obtained, upon using the pL-e6 conjugate, resulting in an enhanced photodynamic effect.

Next, Bengang Xing et al. reported a strategy for fluorescent imaging and photodynamic inactivation of Gram-negative bacteria that was based on the bioconjugation of protoporhyrin IX (PpIX) with an antimicrobial peptide YI13WF, which is known to bind lipopolysaccharides (LPS) [239]. With this bioconjugate, they were able to effectively deliver the PS to the surface of Gram-negative bacterial strains. In addition, there was also an enhancement in the local concentration of the PS due to the effects of multivalency which was a direct effect of the binding affinity of the peptide sequence towards the LPS components. Within the same year, Jean-Philippe Pellois et al. and Marina Gobbo et al. also developed PS conjugates that make use of antimicrobial peptides as agents that can specifically target PSs to bacteria [240,241]. Many more conjugates were developed in the following years and an overview is given in Table 2. It is also worth mentioning that researchers are now focused on overcoming the limitations associated with AMPs (e.g., short plasma half-life). This can be achieved by the incorporation of unnatural amino acids, cyclizing the sequence or modulating the length of the AMP. Additionally, the use of peptoids or N-substituted glycine oligomers have recently been reported and studied by Jiwon Seo et al. [237].

The second group of peptide–PS conjugates is the one obtained by conjugating the PS to CPPs. This group is particularly useful in the context of photochemical internalization (PCI), first described by Berg and co-workers [256]. PCI is a promising technique, which involves the release of endocytosed macromolecules into the cytoplasmic matrix using PSs and light. This avoids degradation of the endocytosed molecules in lysosomes and is therefore regarded as an interesting tool in drug delivery applications.

Within the last 20 years, several researchers have demonstrated that CPPs could be used as PCI agents since TAT or (poly-arginine) R9 labeled with fluorescein, Alexa fluors, tetramethylrhodamine and Cy3 are able to lyse endosomes and deliver nucleic acids and proteins to the cytosol of live cells successfully [257,258]. Remarkably, under “normal” conditions (in the absence of CPP), these fluorophores are innocuous. This indicates that the CPP acts in synergy to elicit photolysis. However, the mechanism of action and how the structure of fluorophore–CPP conjugates impacts this synergistic activity still needs to be unraveled [259]. Though the typically used PSs for PDT are often unsuitable for PCI (due to their non-selective partition to other cellular organelles), some of them, especially the amphiphilic ones, can be used as long as they can be localized to the endolysosomal compartment. Meso-tetraphenyl porphyrindisulfonate, disulfonated aluminum phthalocyanine and disulfonated tetraphenyl chlorin are some examples thereof.

In 2017, A. J. MacRobert et al. hypothesized that the attachment of a conventional photosensitizer to a suitable CPP peptide sequence should result in a conjugate that is able to localize in the lipid bilayer of the endosomal membranes to deliver selective oxidative damage [260]. The hydrophilic protonated peptide will reside at the membrane-aqueous interface whereas the aromatic PS macrocycle will reside in the lipid bilayer. By conjugating a porphyrine derivative (ce6) to the TAT sequence or other positively charged CPP’s, they were able to demonstrate the previously mentioned hypothesis. T. Ohtsuki et al. [261] recently wrote a good review that covers this topic in more detail,

In order to improve the success rate of developing CPP-cargo–PS conjugates, T. Ohtsuki et al. investigated the influence of the linker between the cargo and PS on PCI. They concluded that the PhePhe or LeuLeu linker should be the first choice in future designs [262].

Finally, the group of growth factor receptor targeting peptide–PS conjugates is of great importance in the field of targeted photodynamic therapy. Via specifically targeting malignant tissues, the PS accumulation in normal tissue will be diminished, resulting ideally in less skin photosensitivity. This type of conjugates was first described by M. Barberi-Heyob [263]. Within their study, they coupled the heptapeptide (ATWLPPR), which is known to bind to neuropilin-1 (NRP-1), via a spacer (6-aminohexanoic acid) to 5-(4-carboxyphenyl)-10,15,20-triphenyl-chlorin (TPC), in order to improve TPC delivery to human umbilical vein endothelial cells (HUVEC). Their results demonstrate that TPC-Ahx-ATWLPPR is a more potent PS than TPC in HUVEC expressing NRP-1 [264,265,266].

In 2016, M. G. H. Vicente and co-workers, reported the synthesis of four mesoporphyrin IXpeptide conjugates, based on the LARLLT and GYHWYGYTPQNVI sequences. These conjugates were designed to target EGFR, which is over-expressed in colorectal and other cancers [267]. Another study, carried out by P. C. Lo, employed an EGFR binding peptide (GE11, YHWYGYTPQNVI) linked to Zinc(II) phtalocyanine [268]. The most recent study in this context made use of Gonadotropin-releasing hormone receptor targeting peptides that were conjugated to Protoporphyrin IX. With these conjugates, they aim to obtain a more targeted therapy towards head and neck squamous cell carcinomas [269]. Next to these more conventional type of structures, nanoparticles and liposomes were also modified with growth factor targeting peptides, to enable a more targeted and controlled release of the PS [270,271,272].

Additionally, peptides conjugated to a PS, have also been utilized to target other cellular compartments. A few examples of BODIPY photosensitizer conjugates are listed below. In 2013, a short NLS peptide in connection with a BODIPY photosensitizer was applied by P. Verwilst et al. in order to obtain nuclear localization [273]. In 2009, the Burgess group proved the utility of a bivalent IYIY ligand in connection with a BODIPY PS to target the Tropomyosin Receptor Kinase C (TrkC), a receptor responsible for regulation of neurons and also playing a role in several cancer types (glioblastoma, neuroblastoma) [274]. C. S. Kue et al. also illustrated the immune-stimulatory effect of the IYIY ligand. Thus, the IYIY-I_2_-BODIPY conjugate is not only a targeted PDT agent, but also proved to execute immunomodulation and antitumor activity before photoactivation of the PS, acting as a chemotherapeutic agent [275]. In 2020, S. Y. Ng et al. reported another bivalent ligand (IKIK) connected to a BODIPY, suggesting to actively target the TrkC receptor [276].

#### 4.2.2. Antibody Conjugates

Antibody–PS conjugates offer a new weapon in the arsenal of cancer fighting agents. Whereas in conventional PDT, the PS is also taken up by healthy tissues, targeted phototherapy can be achieved when the monoclonal antibody (mAB) is conjugated to the PS (see Figure 18). This type of strategy is referred to as photo-immunotherapy (PIT) and was demonstrated for the first time in vivo by J. G. Levy and co-workers 1983 [277]. They conjugated hematoporphyrin to a mAB with specificity for the M-1 myosarcoma of DBA/2J mice and were able to inhibit tumor growth. Two years later, they published another paper in which they described the cell killing effect of the CAMAL-1-hematoporphyrin conjugate at concentrations that would not be feasible when hematoporphyrin would have been administered as such [278]. In the first few years that followed, researchers explored the degree of substitution on the mAB and different linking methods [279,280]. Currently, several reviews concerning this topic were published in the last two years and clinical trials are ongoing [281,282,283,284,285]. Given that these recent reviews cover the topic in detail, antibody–PS conjugates will not be covered in more detail in this particular review.

Though these antibody–PS conjugates are very valuable systems in the context of PIT, they do have the disadvantage of having a long half-life time which results in an elevated risk of dark toxicity and phototoxicity in light-exposed skin [287]. Reducing the circulation time is thus of great importance. Therefore, new PS-conjugate constructs have been synthesized that made use of antibody fragments [288,289,290].

The use of small-sized recombinant lama antibodies, also called nanobodies (NB), was reported for the first time in 2014 by S. Oliveira and co-workers [291,292]. Recently, a potential application of NB-PS constructs in oncological animal patients was described [293,294,295]. X. J. Peng et al. presented another photo-immunoconjugate that consists of an anti-EGFR nanobody and benzophenothiazine as PS which can be used for hypoxia resistant photo-immunotherapy [296].

#### 4.2.3. Oligonucleotide (and Their Analogues) Complexes and Conjugates

As illustrated above in Section 2.2, oxidative damage to DNA and RNA strands induced by an uncontrolled ^1^O_2_ production can be a limiting factor for all those strategies (labelling, biosensing, cross-linking) relying on the generation of this oxygen reactive species. Therefore, one must take precautions in order to limit the generalized damage to the payload, especially when the integrity of the target is required. The most logical choice to confine damage is exploiting the natural base-pair recognition of nucleic acids, to confine ^1^O_2_ production locally (Figure 19A). An example can be found in the use of synthetic probes complementary to an oligonucleotide target, decorated with a PS. A valuable example is reported by the Yavin group (see above Section 3.2), in which PNA probes, decorated with RB, could effectively and selectively target the DNA of choice [185]. Several methodologies featuring the conjugation of PS to oligonucleotides and analogues were reported in the literature, exploiting the principle of selective recognition to limit ROS generation to the targeted strand, leading to high levels of selectivity. This was successfully applied to miRNA targeting and imaging [297]; specific targeting through the incorporation of a PS into aptamers [298,299]; realization of DNA cross-linking probes [174]; and self-assembled PNA–Ps conjugates [300]. A similar strategy was explored in our group, where DNA–MB conjugates were used to perform an oligonucleotide-templated, furan-mediated CL reaction. In this case, the use of two separate strands, one bearing furan and the other one bearing the required PS, both adjacently hybridized to the target strand, effectively leads to formation of ICL products. On the other hand, the introduction of both moieties on the same probe, led to the degradation of the targeting probe, with no ICL formation [175,301].

In an alternative approach, molecular beacon probes were successfully developed for selectively targeting oligonucleotides, displaying a turn-on ^1^O_2_-production in the presence of the complementary target, taking advantage of the quenching effect when the probes are not hybridized to their natural target [302,303,304,305,306,307] (Figure 19B), due to the presence of a quencher. The use of these molecular beacons has several advantages over the use of regular probes, since the ^1^O_2_ generation can be selectively induced, only in the presence of a complementary target, able to open the hairpin structure and separate the PS from its quencher, thus lowering side effects related to an excessive ROS generation. An elegant solution to tailor ^1^O_2_ generation was reported by Yang and co-workers, who took advantage of the increased expression of diaza-reductase enzyme in cells under hypoxia conditions. They reported on a DNA nanostructure consisting of oligonucleotide sequences bearing a cyanine dye, a black-hole quencher linked through an aza-bridge and a G-rich sequence folding into a G-quadruplex (G4), able to carry a G4-binding porphyrin (TMPyP4) as PS. Due to the presence of the quencher, the cyanine fluorescence and the ^1^O_2_ production of the PS are limited. However, when the nanoconstruct is internalized, the aza-reductase expressed by hypoxic cells cleaves off the quencher, thus restoring the properties of the dyes, enabling a selective PDT approach [308].

Another solution to limit uncontrolled ROS generation can consist in the incorporation of milder PSs on the targeting probes. In this context, we reported on the incorporation of TAMRA on PNAs, to tailor the activation of a furan pro-reactive moiety in a templated fashion. This allowed performing an oligonucleotide-templated PNA photo-ligation with no collateral damage on the biotin tag installed on the reporter probe [179]. Even if only a very limited amount of ^1^O_2_ is generated when using such mild PSs, the fact that it is generated in close proximity of the target largely compensates for that, thus creating carefully balanced conditions allowing the avoidance of collateral damage.

A last but effective strategy to localize ^1^O_2_ production, is using PSs able to bind certain targeted DNA structures. Some very recent examples are reported by the group of Linker, who reported on the use of pyridinium alkynylanthracenes as PSs featuring an enhanced ^1^O_2_ generation when binding double stranded DNA structures, in presence of green light, which were successfully applied in the context of PDT [309,310]. Besides the known affinity of certain PSs to dsDNA and RNA, some of the reported dyes exhibit preferences in binding alternative secondary structures, such as G4s (Figure 19C), and they can therefore be used to confine ^1^O_2_ generation, increasing their potential therapeutic effect [311]. This concept was successfully exploited for increasing the selectivity of RAS-directed PDT, exploiting G4-binding porphyrins, which can stack onto the G4-DNA target and preferentially direct the light-induced ^1^O_2_ production to those targeted cells in which RAS is overexpressed [312,313]. Several G4-binding PSs, mostly featuring porphyrins and phthalocyanines, were exploited in this context, as described in the review by Kawauchi et al. [314] As discussed above, our group reported on the use of MB as a G4-binding PS, in order to mediate red light-triggered G4-alkylation, exploiting the simultaneous localization of PS and pro-reactive ligand on the same targeted structure. Despite the high quantum yield of MB, the proximity effect induced by the co-localization enabled an efficient alkylation of the target at very low PS concentration, ensuring fast and quantitative activation of the system even in the presence of competing dsDNA [176]. The high level of selectivity that can be reached by using this ^1^O_2_-based strategy could in principle allow to promote the sequence-specific alkylation of the desired structure [315].

**Figure 19 molecules-27-00778-f019:**
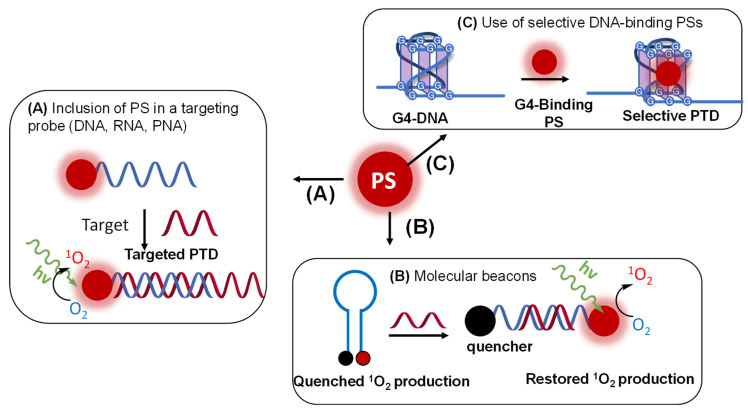
Tailoring ^1^O_2_ generation with DNA and their analogues. (**A**) Approaches based on the inclusion of a PS on a synthetic probe complementary to a desired target [181]. (**B**) Turn-on approaches based on molecular beacons target [302,303,304,305,306,307]. (**C**) Use of selective DNA binding [312,313].

## 5. Conclusions

^1^O_2_, present in vivo as a ROS species, is conveniently produced upon the irradiation of a PS with light of a suitable wavelength. In the past few years, the interest in the photosensitized production of ^1^O_2_ as an oxidation strategy rose tremendously in various fields of research, ranging from organic synthesis to various high-end biomedical applications. This review highlights the use of singlet oxygen in various biochemical labelling, cross-linking and biosensing methodologies. However, the application of ^1^O_2_ in a biological context should be planned with care, as it can cause damage to various vital cellular compounds, such as DNA, proteins, cholesterol and membrane phospholipids. Therefore, considerable research efforts were directed towards minimizing oxidative damage, requiring intensive protocol optimization (type and concentration of PS, light intensity, etc.) and more localized and targeted productions of ^1^O_2_. In the case of the latter, various methodologies were developed, such as the genetically encoded PSs (miniSOG) and PS conjugates. More recently, research was also directed to the use of nanoparticles in combination with PSs in order to generate ^1^O_2_ locally. As the role of ^1^O_2_ in the development of various diseases becomes more and more clear, we foresee future research efforts to be directed towards tailored and localized production approaches, in particular through the use of photosensitized nanoparticles.

## Figures and Tables

**Figure 1 molecules-27-00778-f001:**
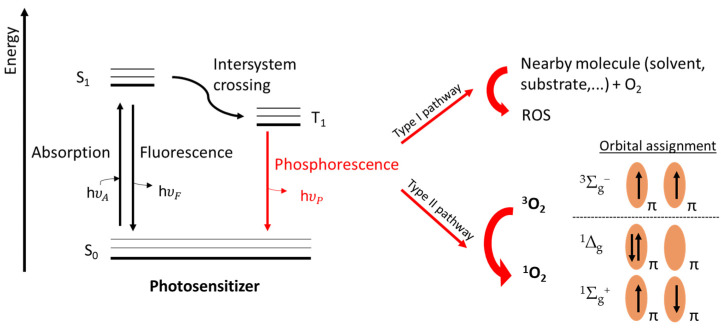
Jablonski diagram and light-dependent photo-oxidation mechanisms (type I and II) [17].

**Figure 2 molecules-27-00778-f002:**
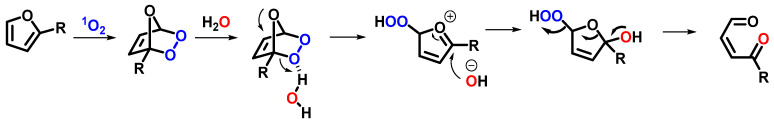
Reaction mechanism of furan oxidation with ^1^O_2_ in aqueous environment.

**Figure 3 molecules-27-00778-f003:**
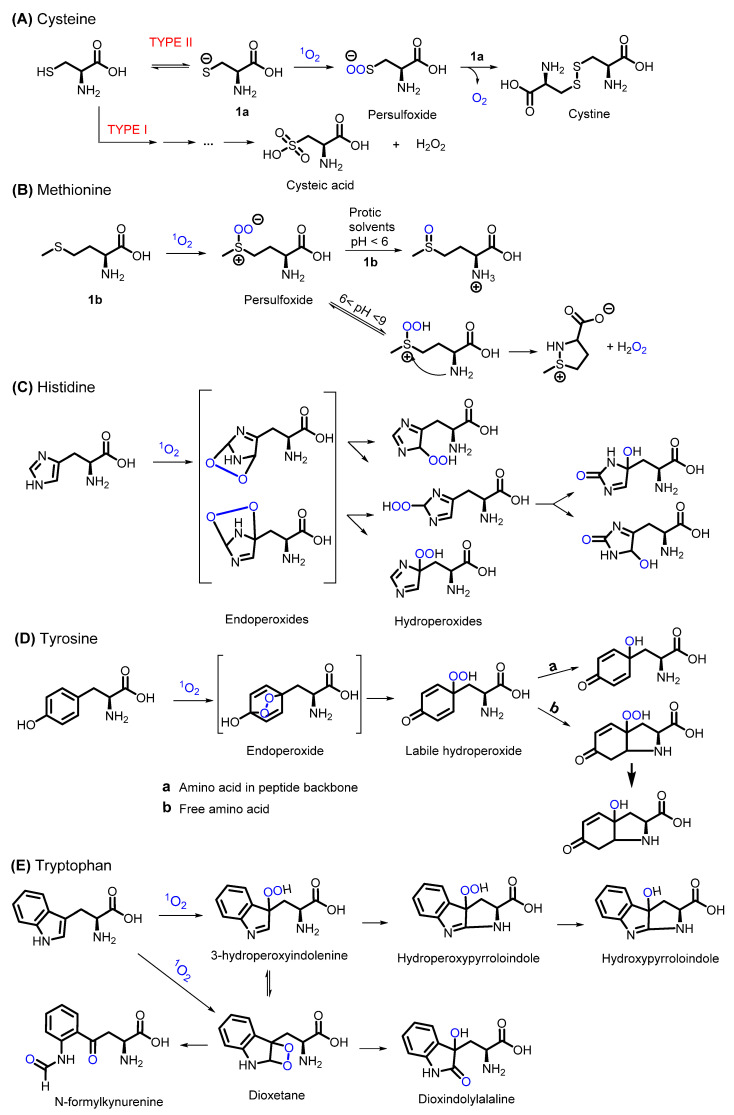
(**A**) Type I and II oxidation pathways of cysteine in the presence of ^1^O_2_, (**B**) ^1^O_2_-mediated oxidation cascade of methionine, (**C**) Peroxide formation on histidine in the presence of ^1^O_2_, (**D**) ^1^O_2_-mediated damage onto tyrosine, (**E**) Oxidative damage onto tryptophan due to the presence of ^1^O_2_.

**Figure 4 molecules-27-00778-f004:**
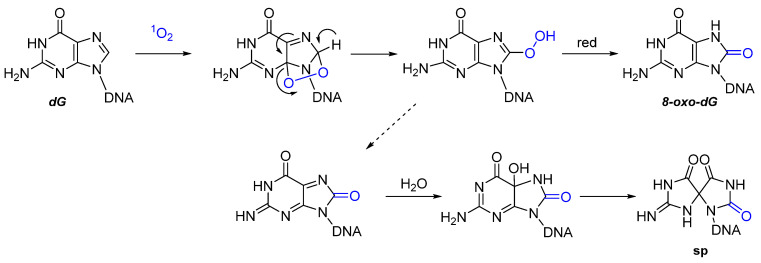
^1^O_2_-mediated oxidation pathway of dG, resulting in the formation of 8-oxo-dG and spiroiminodihydantoin derivative (sp).

**Figure 5 molecules-27-00778-f005:**
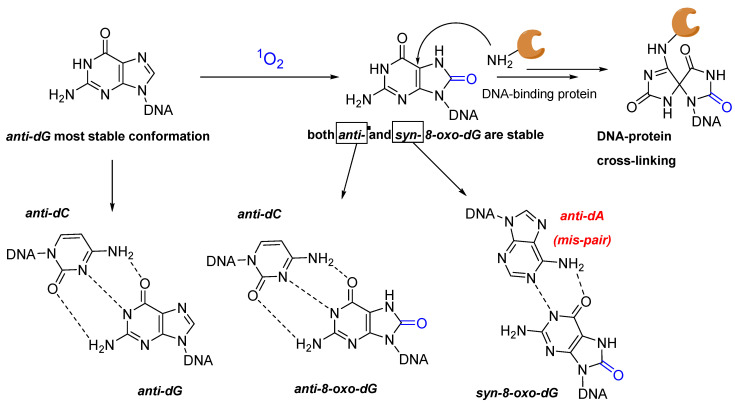
DNA mispairing and CL formation with DNA-binding proteins upon the ^1^O_2_-mediated oxidation of dG.

**Figure 6 molecules-27-00778-f006:**
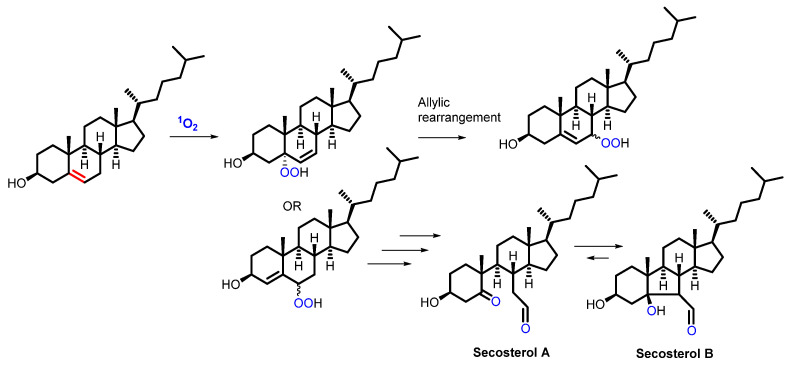
General overview of ^1^O_2_ oxidation products of cholesterol, illustrated in [25].

**Figure 7 molecules-27-00778-f007:**
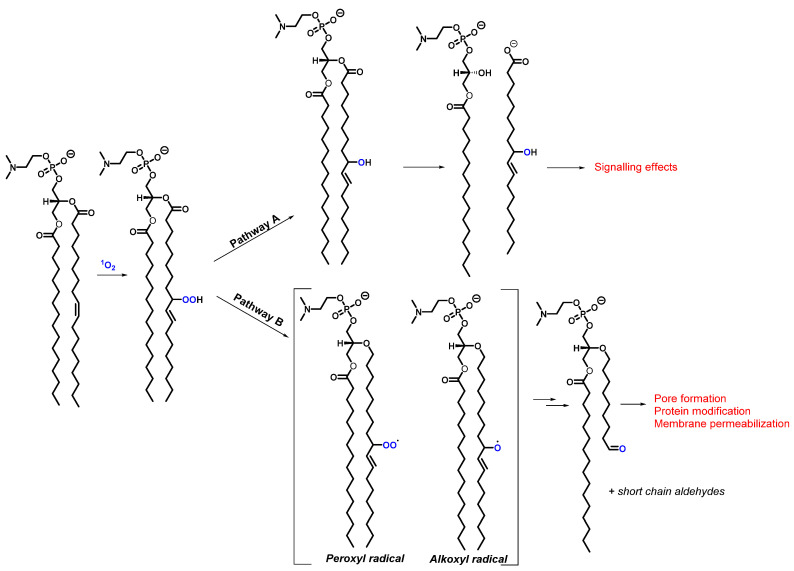
Membrane phospholipid ^1^O_2_ oxidation and decomposition pathways, illustrated in [25].

**Figure 8 molecules-27-00778-f008:**
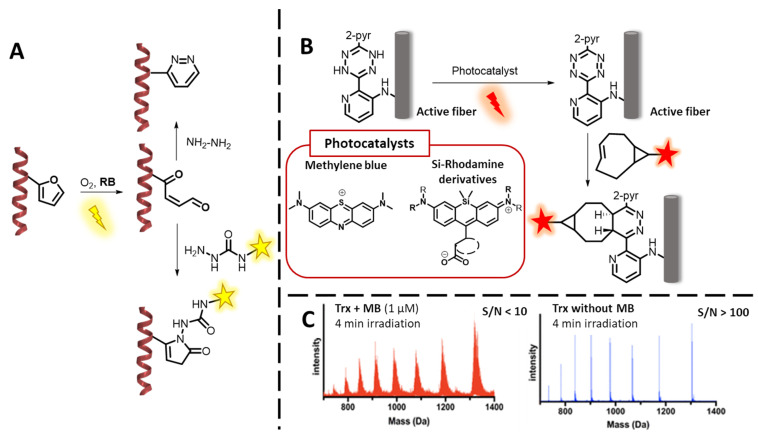
(**A**) ^1^O_2_-mediated oxidation of furan in a bio-ligation method, shown [106]. (**B**) Turn-on tetrazine ligation method, introduced in [107]. (**C**) ESI mass spectrometry illustration of the oxidative damage to the protein thioredoxin (Trx) caused by the presence of MB and irradiation, used with permission from [108]. Copyright 2021 American Chemical Society.

**Figure 9 molecules-27-00778-f009:**
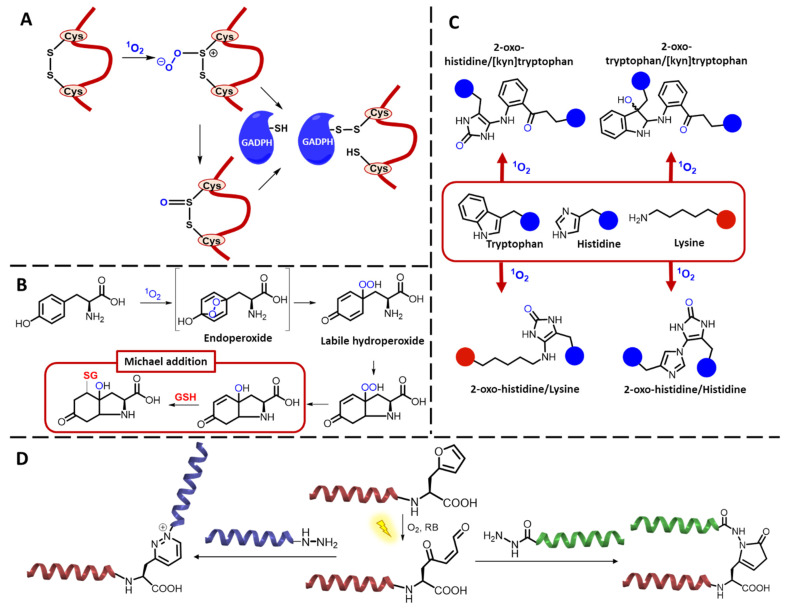
(**A**) Possible mechanistic pathway for the formation of new intermolecular disulphide bridge upon the presence of ^1^O_2_, introduced by S. Jiang et al. in [47]. (**B**) Proposed mechanistic pathway for the glutathione conjugation onto oxidized tyrosine residues in peptides and proteins, further elaborated in [129]. (**C**) Proposed intermolecular lysozyme cross-links after ^1^O_2_-mediated oxidation of histidine and tryptophan, introduced in [46]. (**D**) ^1^O_2_-mediated peptide–peptide ligation method, shown in [106].

**Figure 10 molecules-27-00778-f010:**
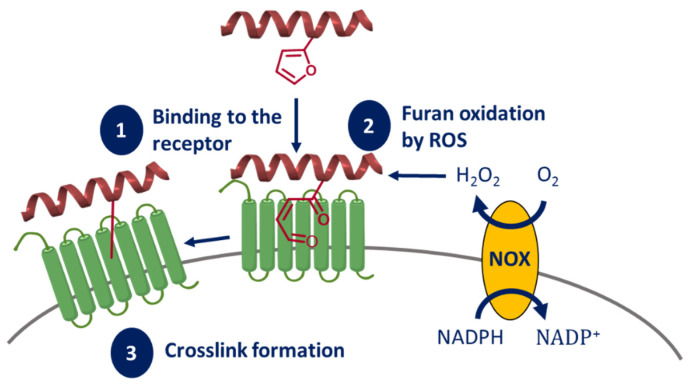
Furan oxidation by ROS produced by the NOX enzymes applied in the CL of a peptide ligand to a receptor, shown in [153].

**Figure 11 molecules-27-00778-f011:**
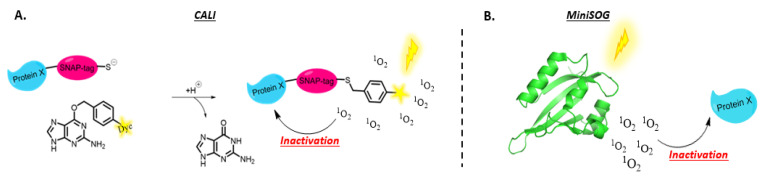
(**A**) Snap-tag fusion strategy shown in [163]. (**B**) CALI based on miniSOG (PDB:6GPV).

**Figure 12 molecules-27-00778-f012:**
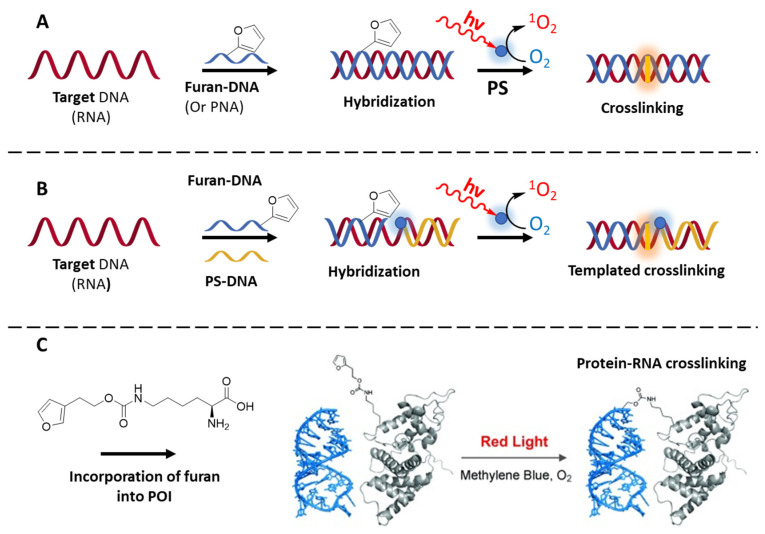
Examples of furan-based CL methodologies relying on ^1^O_2_ photo-induced generation. (**A**) Original methodology proposed by Madder in [172]. (**B**) Follow-up study with templated DNA-CL described by Madder and colleagues in refs [174,175]. (**C**) Protein–RNA CL upon incorporation of genetically encoded furan moiety. Described by Summerer in [180].

**Figure 13 molecules-27-00778-f013:**
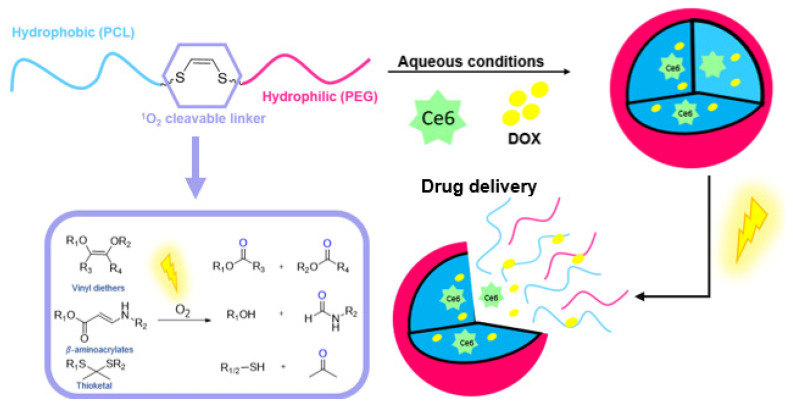
Vinyldithioether linker used in a PCL and PEG containing block copolymer. In aqueous conditions, micelles are formed. Upon light irradiation, ^1^O_2_ is produced (due to the presence of Ce6) which results in the destruction of the micelle and thus the release of the drug, shown in [196]. Other ^1^O_2_ cleavable linkers are shown in the figure as well.

**Figure 14 molecules-27-00778-f014:**
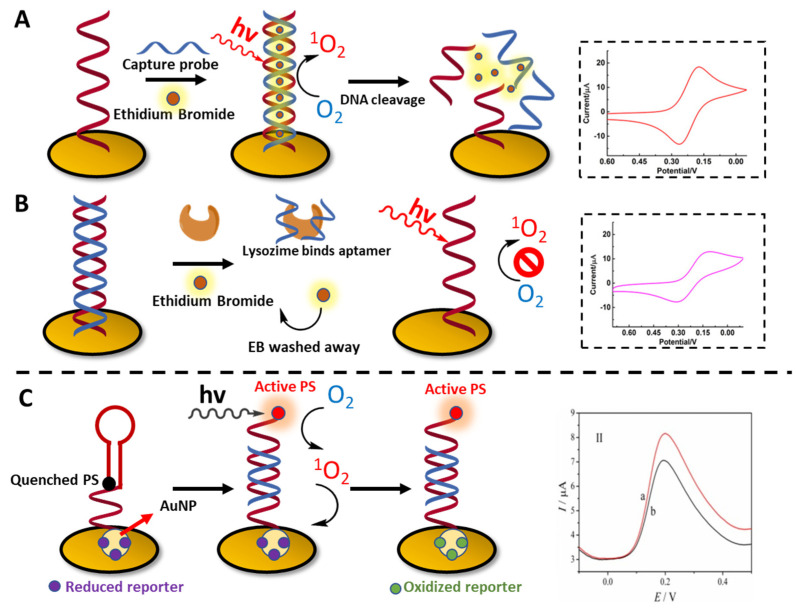
Electrochemical detection methodologies based on ^1^O_2_ generation, in the presence of DNA probes. (**A**) Detection of BRCA1 gene by cyclic voltammetry, as proposed by Zhang in [209]. (**B**) Detection of lysozyme proposed in ref. [210]. (**C**) Cyclic voltammetry-based detection of DNA, showed in ref. [211].

**Figure 16 molecules-27-00778-f016:**
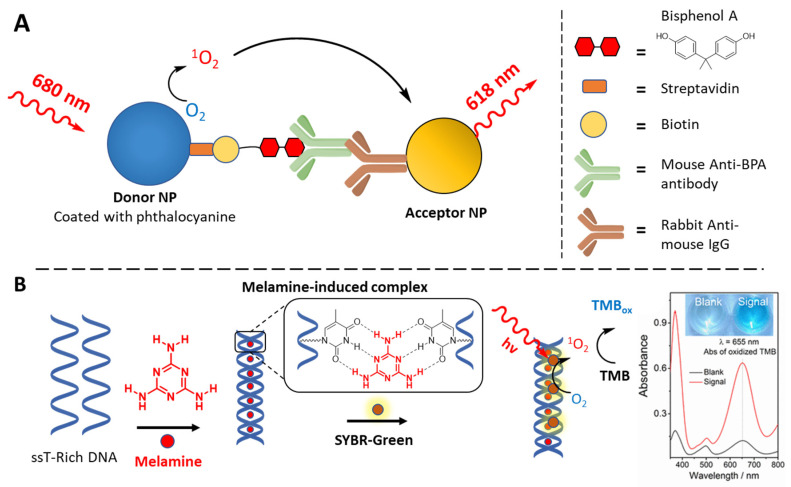
Approaches for the detection of small molecules based on ^1^O_2_ generation. (**A**) Bisphenol. A detection proposed by Guo and co-workers in [221]. (**B**) Colorimetric melamine detection based on the hybridization between two T-rich DNA strands and SYBR-Green intercalation, described in ref. [222].

**Figure 17 molecules-27-00778-f017:**
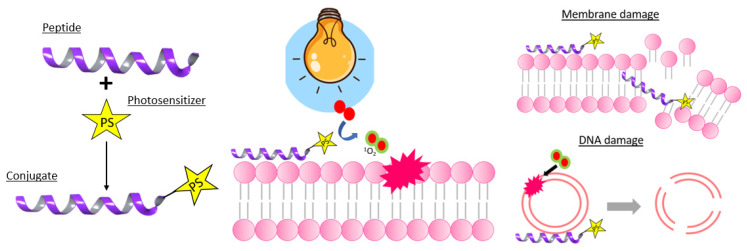
Representation of the mechanism of action of PS–AMP conjugates. Light irradiation of the conjugate damages the bacterial membrane, which results in cleavage of DNA, shown in [237].

**Figure 18 molecules-27-00778-f018:**
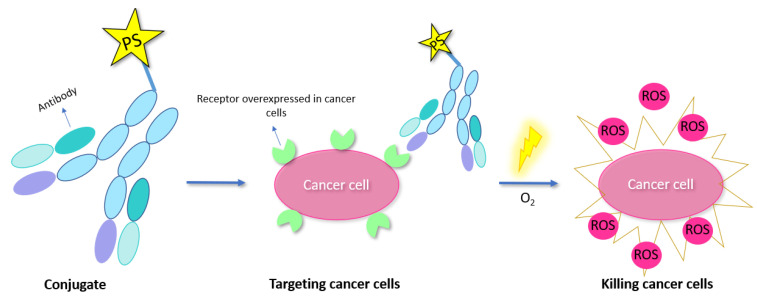
Representation of the mechanism of action of antibody–PS conjugates. Upon light irradiation of the conjugate, ROS species are generated which results in killing of the cancer cells, shown in [286].

**Table 1 molecules-27-00778-t001:** Types of PSs [8,21,22].

Classes	Examples
Organic dyes and aromatic hydrocarbons	Rose Bengal (RB), methylene blue (MB), quinones
Tetrapyrroles	Porphyrins, Phtalocyanines
Transition metal complexes	Metal complexes of Ruthenium (Ru(II) tris-bipyridine), Pt- and Pd(II) mixed ligands
Semiconductors	TiO_2_, ZnO
Inorganic nanoparticles	(CdSe, CdSe/ZnS) quantum dots, porous silica nanocrystals, gold, silver, platinum and nichel nanoconstructs (nanoparticles, nanoclusters, nanorods)

**Table 2 molecules-27-00778-t002:** Overview of developed AMP–PS conjugates.

Antimicrobial Peptides	PS	References
KLA(KLAKLAK)_2_	Eosin Y	[241,242,243]
YVLWKRKRKFCFI-NH_2_	protophophyrin IX	[244]
(GKRWWKWWRR)_2_KGGK	Chlorin e6	[244]
Polymyxin B	Chlorin e6, Porphyrins	[245,246]
GGGKKKKKRWRWRW	Phthalocyanine	[247,248]
Cyclic Bactenicin		
Buforin II	[Ru(bpy)_3_]^2+^	[249]
Poly-L-Lysine	Chlorine e6, BOHTMPn, GlamTMPn	[238,250]
Aurein 1.2	Methylene blue, Chlorin e6, Curcumin	[251]
epsilon-poly-Lysine	CPZ	[252]
Peptoid	TMPyp	[237]
Apidaecin	Porphyrins	[240,245,246,248,253,254,255]
WRF		
RWRW		
FRWWRR		
Polymyxin Magainin Buforin		

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
