# Peer review of "A Photosensitized Singlet Oxygen (1O2) Toolbox for Bio-Organic Applications: Tailoring 1O2 Generation for DNA and Protein Labelling, Targeting and Biosensing"

_molecules, 2022, doi:10.3390/molecules27030778_

Round 1

Reviewer 1 Report

This review gives a wide spreading overview of applications of singlet oxygen in the field of bio-organic chemistry. It includes reactions of peptides, nucleic acids, carbohydrates and lipids. The next section describes specific applications such as protein labeling, cross-linking, desactivation or sensing using 1O2 reactions. In the final section methods are presented, where the generation of 1O2 is controllable by using bimolecular interaction.

 I can recommend this manuscript for publication because it tells exciting aspects of this chemistry and guarantees a thorough survey.

The text is well written and only some minor aspects are worth for correction:

  1. Figure 1: Is really 1Σ g+ the desired character group for 1O2 or rather 1Δg ?
  2. Explain the abbreviation Trx
  3. Page 9, line 285 explain “ICL” (CL=cross-linking)
  4. Compare Page 10 bottom versus Figure 7: membrane “permeabilization”
  5. Page 27 Table and Figure 16. Correct “Phthalocyanine”

Finally, some very recent publications on alkynyl pyridyl anthracenes and their singlet oxygen reactions with DNA should be cited in this review (https://doi.org/10.1002/chem.202101918 and DOI: 10.1111/php.13554)

Author Response

We thank the Editor for handling this submission and for giving us the chance to submit this manuscript to the journal. Following the indications given, we prepared a revised version of the manuscript, in which we try to meet all suggestions and points raised by the reviewers. The changes made to the manuscript were highlighted in yellow in the revised version of the document.

Reviewer 1 comments:

This review gives a wide spreading overview of applications of singlet oxygen in the field of bio-organic chemistry. It includes reactions of peptides, nucleic acids, carbohydrates and lipids. The next section describes specific applications such as protein labeling, cross-linking, desactivation or sensing using 1O2 reactions. In the final section methods are presented, where the generation of 1O2 is controllable by using bimolecular interaction.

I can recommend this manuscript for publication because it tells exciting aspects of this chemistry and guarantees a thorough survey.

The text is well written and only some minor aspects are worth for correction:

Response to reviewer 1 comments:

We thank the reviewer for appreciating our review work and for the recommendation for publication.

  1. Figure 1: Is really 1Σ gthe desired character group for 1O2 or rather 1Δg ?

We thank the reviewer for pointing out the typo in the figure, which indeed was not including the orbital assignment for 1Δg. The figure was corrected in the final version of the manuscript (please refer to page 2, line 62 of the revised document).

2. Explain the abbreviation Trx

Following the reviewer’s suggestion, the abbreviation trx (thioredoxin) was now explained in the caption of figure 8 (please refer to page 13, line 404 of the corrected version) as well as in the main text of the document (please refer to page 14, line 416 of the corrected version).

3. Page 9, line 285 explain “ICL” (CL=cross-linking)

Following the reviewer’s suggestion, the abbreviation used for interstrand-crosslinking was explained in the main text of the document (please refer to page 10, line 290 of the corrected version).

4. Compare Page 10 bottom versus Figure 7: membrane “permeabilization”

We thank the reviewer for highlighting the difference in writing style (American vs British English) of the word ‘permeabilization’. The term was adapted in Figure 7 (page 12) in order to have an equal writing style in both the figure and the text.

5. Page 27 Table and Figure 16. Correct “Phthalocyanine”

We thank the reviewer for pointing out the typos, which were corrected in the final version of the manuscript (please refer to figure 16, page 25, line 850 and to table 2, page 27 line 940).

Finally, some very recent publications on alkynyl pyridyl anthracenes and their singlet oxygen reactions with DNA should be cited in this review (https://doi.org/10.1002/chem.202101918 and DOI: 10.1111/php.13554).

We thank the reviewer for pointing out these articles, which we have missed during our literature research. Since these works describe the enhanced singlet oxygen production of anthracene dyes when complexed to DNA, these articles were inserted in the last section of the review (4.2.3 “Oligonucleotide (and their analogues) complexes and conjugates”). Please, refer to page 31, lines 1083-86 of the revised version of the document:

“Some very recent examples are reported by the group of Linker, who reported on the use of pyridinium alkynylanthracenes as PSs featuring an enhanced  1O2 generation when binding double stranded DNA structures, in presence of green light, which were successfully applied in the context of PDT.”

Reviewer 2 Report

The present submitted review is very useful coming from a well-established contributor to the organic and bio-organic chemistry field and especially the bioorganic applications of 1O2. The focus is on the production of singlet oxygen and its chemical reactivity with applications on organic synthesis but mainly covers its reactivity in biological systems. The manuscript is very well written and provides all the recent information about the latest findings concerning the biomedical applications of the photosensitized produced singlet oxygen. The detailed studies for the oxidative damage to DNA, proteins, cholesterol and membrane phospholipids by singlet oxygen as one very strong ROS are very well presented and documented. There are many open questions on how to minimize oxidative damage in order to broaden its application in photodynamic therapy among other.

Moreover, the development of various kinds of photosensitizers over the last decade is covered in detail and the readers in the field will find it very informative and of great interest.

The authors are encouraged to include in the introduction the following important references:

The work of Foote and Wexler is considered as the milestone of singlet oxygen research.  They proposed that the lowest excited electronic state of molecular oxygen was the pertinent reaction intermediate in photosensitized oxidations.

1)Foote, C. S. and S. Wexler (1964) J. Am. Chem. Soc. 86, 3879–3880.

2)  Foote, C. S. and S. Wexler (1964) J. Am. Chem. Soc. 86, 3880–3881.

3) Foote, C. S. (1968) Science 162(3857), 963–970.

4) A recent review concerning the discovery and chemistry of singlet oxygen.

Photochemistry and Photobiology 2021, 97, 1182-1228.

Please take in consideration also a few editorial comments:

On p.11, line 365, the numbering is 3.0, and in the Title instead of O2 would it be more suitable 1O2?

The numbering 3.1.11 should become 3.1.1

Line 372: instead of “O2-based peptide…” would it be more suitable 1O2?

Figures 12,14 and 16 are much larger than the rest and the labels and captions inside these schemes do not follow the same or similar settings compared to the rest in the manuscript. For better uniformity these figures can be improved.

A space is missing on line 730 between “featuring” and “1O2”.

Author Response

We thank the Editor for handling this submission and for giving us the chance to submit this manuscript to the journal. Following the indications given, we prepared a revised version of the manuscript, in which we try to meet all suggestions and points raised by the reviewers. The changes made to the manuscript were highlighted in yellow in the revised version of the document.

Reviewer 2 comments:

The present submitted review is very useful coming from a well-established contributor to the organic and bio-organic chemistry field and especially the bioorganic applications of 1O2. The focus is on the production of singlet oxygen and its chemical reactivity with applications on organic synthesis but mainly covers its reactivity in biological systems. The manuscript is very well written and provides all the recent information about the latest findings concerning the biomedical applications of the photosensitized produced singlet oxygen. The detailed studies for the oxidative damage to DNA, proteins, cholesterol and membrane phospholipids by singlet oxygen as one very strong ROS are very well presented and documented. There are many open questions on how to minimize oxidative damage in order to broaden its application in photodynamic therapy among other.

Moreover, the development of various kinds of photosensitizers over the last decade is covered in detail and the readers in the field will find it very informative and of great interest.

We thank the reviewer for appreciating our review work and for the positive comments on the manuscript.

The authors are encouraged to include in the introduction the following important references:

The work of Foote and Wexler is considered as the milestone of singlet oxygen research.  They proposed that the lowest excited electronic state of molecular oxygen was the pertinent reaction intermediate in photosensitized oxidations.

1)Foote, C. S. and S. Wexler (1964) J. Am. Chem. Soc. 86, 3879–3880.

2)  Foote, C. S. and S. Wexler (1964J. Am. Chem. Soc. 86, 3880–3881.

3) Foote, C. S. (1968Science 162(3857), 963–970.

4) A recent review concerning the discovery and chemistry of singlet oxygen.

Photochemistry and Photobiology 2021, 97, 1182-1228.

Asnwers to reviewer 2 comments:

We would like to thank the reviewer for highlighting the missing references concerning the discovery of singlet oxygen as the reaction intermediate in photosensitized oxidation reactions. These references were added in the introduction section in the paragraph that discusses the photosensitized production of singlet oxygen (page 1-2, lines 40-44):

‘In the early 60s, it was proposed by C. S. Foote and S. Wexler that 1O2, the excited state of molecular oxygen with the lowest energy, was the reactive intermediate in photosensitized oxidation reactions. This proved to be a milestone in 1O2 mediated research, paving the way for its use in various (bio-) organic reactions and photodynamic therapy [13-16].’

Please take in consideration also a few editorial comments:

On p.11, line 365, the numbering is 3.0, and in the Title instead of O2 would it be more suitable 1O2?

The numbering 3.1.11 should become 3.1.1                                                                         

Line 372: instead of “O2-based peptide…” would it be more suitablwe 1O2?

We thank the reviewer for pointing out the inconsistencies. Following their suggestion, we adapted the text in the revised version of the manuscript (please refer to page 12, line 369 and 376 of the revised document).

Figures 12,14 and 16 are much larger than the rest and the labels and captions inside these schemes do not follow the same or similar settings compared to the rest in the manuscript. For better uniformity these figures can be improved.

As the reviewer points out, the figures indicated are larger and the labelling slightly different as compared to the other figures of the manuscript. Taking their comment into account, we adapted the figures in the revised manuscript accordingly (please refer to figure 12, page 19, figure 14, page 23 and figure 16, page 25 of the revised document).

A space is missing on line 730 between “featuring” and “1O2

Following the reviewer’s comment, the typo was corrected in the revised version of the manuscript (page 21, line 734).